# PAML: MoE-Based Partitioning and Merging Framework for Solving Large-scale Multi-task VRPs

## Abstract

The Vehicle Routing Problem (VRP) serves as a fundamental optimization problem in modern logistics and supply chain management, where efficient solutions to its large-scale multi-task variants are crucial for reducing transportation costs and improving resource allocation efficiency. Although significant progress has been made in intelligent solving approaches for small- and medium-scale VRPs, current methods still face three major limitations when dealing with real-world large-scale multi-task scenarios: 1) Neural heuristic models trained on small-scale datasets struggle to generalize effectively to larger problem instances; 2) The computation time of traditional optimizers grows nonlinearly with problem scale, making them impractical for real-time decision-making; 3) Current solution approaches lack systematic mechanisms to handle the complex interactions and constraints between multiple concurrent tasks in an integrated manner. To address these challenges, this paper proposes the MoE-Based Partitioning and Merging (PAML) framework, with two key innovations: 1) A learnable and scalable implicit partitioner capable of handling multiple VRP variants, which optimizes partitioning strategies through end-to-end reinforcement learning, effectively overcoming training data scale limitations; 2) A dynamic merging mechanism based on polar angle clustering that enables intelligent control of subproblem sizes. This design allows efficient parallel solving of the partitioned VRP subproblems. Experimental results demonstrate that across various synthetic and real-world multi-task VRP variants of different scales, the PAML method shows remarkable improvements over its base solver model: reducing route length by up to 48.71% for 2000-node problems and 20.66% for 1000-node problems. For real-world CVRPLIB instances, PAML achieves a 16.78% reduction in routing distance compared to Multi-Task Vehicle Routing Solver with MoE (MVMoE) while delivering comparable performance to OR-Tools. Remarkably, PAML requires only one-tenth of OR-Tools' computation time (0.95s vs 14.23s on average).

## 1 Introduction

The Vehicle Routing Problem (VRP), first proposed by Dantzig & Ramser (1959), is a fundamental combinatorial optimization problem in logistics and supply chain management. With increasing complexity of practical scenarios, VRP and its variants (CVRP, VRPTW, VRPB) have become critically important research topics in operations research and artificial intelligence due to their NP-hard characteristics. Traditional optimization methods, including exact algorithms and heuristic/metaheuristic approaches, achieve good results for small-scale VRP instances (Aarts & Jan Karel Lenstra, 2003; Naddef & Rinaldi, 2001; Cordeau et al., 2002), but lack generalization and efficiency in large-scale, multi-task scenarios. In recent years, AI-driven approaches have emerged, with neural heuristics offering end-to-end learning and good generalization. Notable advances include Graph Convolutional Neural Networks (Gasse et al., 2019), attention-based models (Kool et al., 2019), and multi-task architectures like MoE models (Zhou et al., 2024; Shazeer et al., 2017). For detailed reviews and evaluations, see Section Related Work.

Despite these advances, current methods still face significant challenges in large-scale multi-task scenarios: **limited multi-task optimization** capabilities across diverse VRP variants with varying

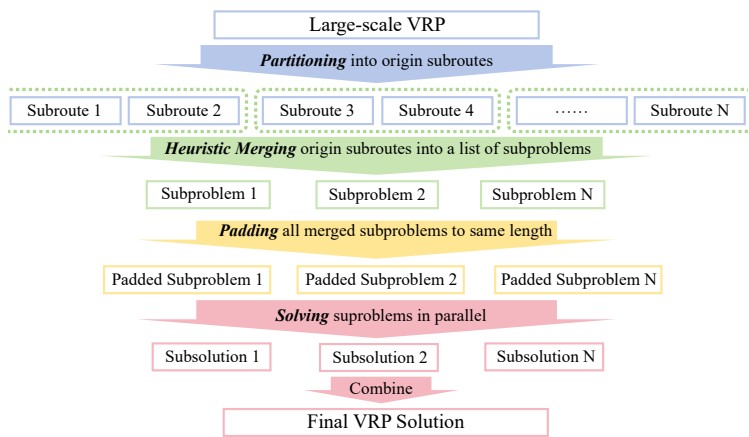

Figure 1: Our PAML framework. In the first phase (blue), the learning model splits the large VRP into multiple small primitive subroutes while preserving the original constraints of the VRP, such as capacity, maximum vehicle route length, backhaul demand, etc. Then, in the second stage (green), the primitive subroutes are merged into subproblems according to the constraints. In the third stage (yellow), all merged subproblems are padded to the same number of nodes. In the fourth stage (red), all padded subproblems are solved in parallel.

constraints, **computational inefficiency** for real-time applications due to lengthy solving times, and **poor generalization** from small training instances to large-scale problems due to insufficient global structural understanding.

Given these limitations, we propose: Can we develop an AI-based solving method that adapts to multi-task optimization, maintains high computational efficiency, and demonstrates stronger generalization capabilities? We attempt to achieve this through two key approaches: 1) **Subproblem Partition:** To enhance generalization for large-scale multi-task VRPs, we adopt a partitioning strategy. Inspired by the two-stage division approach of the Two-stage Dividing Method (TAM) (Hou et al., 2023), we decompose a large problem into smaller, manageable multi-task subproblems, enabling parallel and more effective solving. 2) **Mixture-of-Experts (MoE) Solving:** To address the challenges of multi-task optimization, we leverage a Mixture-of-Experts (MoE) architecture. Drawing inspiration from the MVMoE model by Zhou et al. (2024), we use its ability to adaptively handle diverse VRP variants and constraints to solve the partitioned multi-task subproblems.

Based on these ideas, we propose the MoE-Based Partitioning and Merging (PAML) framework. Using a divide-and-conquer strategy, we intelligently decompose large-scale multi-task VRP into smaller subproblems for parallel solving through an MoE-based partitioner. During development, we innovatively introduce a dynamic merging mechanism for intelligent subproblem scale regulation. The method consists of three stages: 1) **Partition:** PAML first employs a deep neural network model based on the MoE architecture to partition VRP nodes into a list of initial subproblems. This model is data-driven and adaptively learned, requiring no preset rules. It is trained end-to-end via reinforcement learning (REINFORCE algorithm) to maximize subsequent solving rewards, with a Greedy Rollout Baseline for training stability. 2) **Merging:** For the divided initial subproblem sequence, PAML then applies heuristic merging strategies based on geometric information (centroids, polar angles relative to depot), merging subproblems according to preset parameters (fixed quantity or target node count). Through this merging process, PAML can dynamically regulate to optimize subproblem scale and quantity, balancing the divided subproblems' ability to preserve global information with their solving complexity. 3) **Parallel Solving:** Finally, the merged subproblems can be solved in parallel using either traditional or AI methods.

The primary contributions of this project include: 1) We developed a **Novel Three-Stage Partition-Merge-and-Solve Framework** specifically designed for large-scale, multi-variant VRPs. It effectively combines deep learning-based implicit partitioning with high-performance multi-task subproblem solving mechanisms, enhancing the capability to solve complex VRP instances. 2) We trained an **Intelligent MoE-based Implicit Partitioner** that has been applied to the task of implicit VRP partitioning. This partitioner, trained via end-to-end reinforcement learning, can adaptively

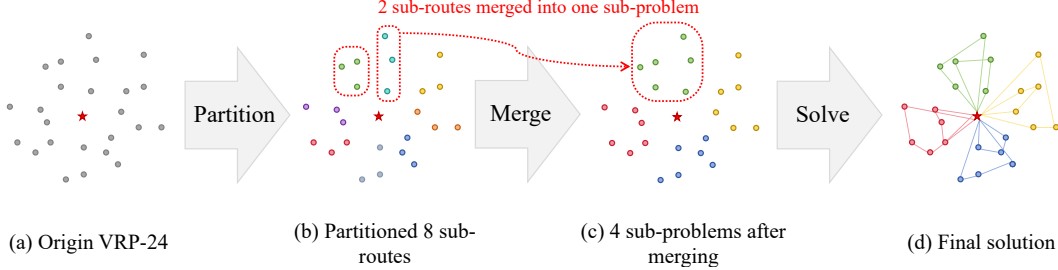

(a) Origin VRP-24     (b) Partitioned 8 sub-routes     (c) 4 sub-problems after merging     (d) Final solution

Figure 2: A CVRP sample solved using our PAML method. (a): Origin CVRP instance. (b): Subroutes generated after division. (c): Merged and Padded Subproblems. (d): Final solution.

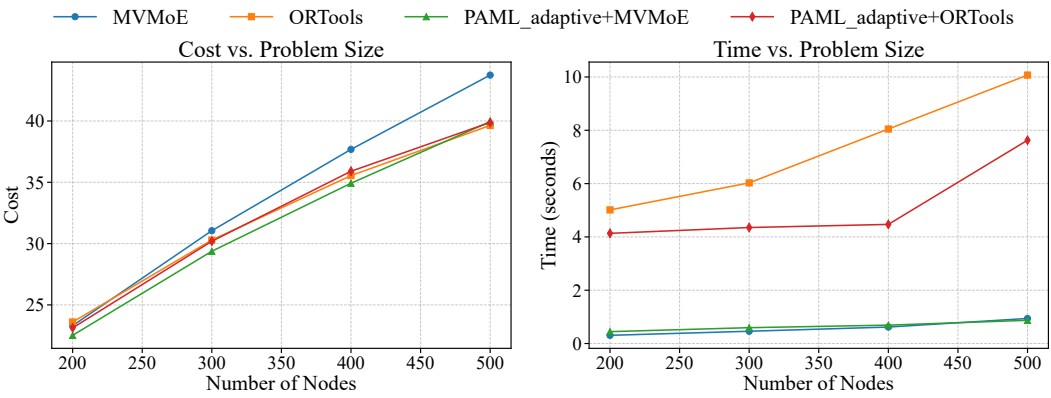

Figure 3: Solving cost (left, smaller is better) and solving time (right, smaller is better) on VRPs of different sizes (200 to 500-nodes) using different methods. We can see that when the size of the VRP is around 200 nodes, the PAML method (PAML_adaptive+MVMoE), which solves the subproblems using the MVMoE solver, outperforms each of the other methods, and also maintains an advantage in terms of solving speed.

generate high-quality partition sequences based on problem characteristics, eliminating the need for manually designed partitioning rules. 3) We propose and develop a **subproblem merging method**. We have thoroughly studied and experimentally compared various geometrically-informed subproblem merging strategies, and finally identified subproblem merging methods for various multi-task VRPs of various sizes, which can dynamically regulate the size of the solved subproblems. This allows for better matching with the preferred working range of subsequent subproblem solvers, thereby improving overall solving efficiency and solution quality. 4) We have conducted systematic experimental evaluations across multiple datasets (including generated benchmarks and real-world public VRPs), with results demonstrating three key advantages: **Multi-task capability** shows robust generalization across diverse VRP variants through consistent solution quality improvements; **Large-scale performance** achieves significant route length reductions of 3.40% (200-node) and 48.71% (1000-2000 nodes) compared to MVMoE Solver, while matching OR-Tools' solution quality on real-world CVRPLIB instances (100-1000 nodes); and **Computational efficiency** maintains just 10% of OR-Tools' processing time for 200-node problems, delivers 91.20% faster computation for 1000-2000 node problems, and realizes 93.32% time reduction (0.95s vs 14.23s) on real-world cases with equivalent solution quality.

## 2 RELATED WORK

### 2.1 TRADITIONAL METHODS

Early research relied on local search methods (Aarts & Jan Karel Lenstra, 2003), which, while simple, often get trapped in local optima, limiting their effectiveness for complex instances. Branch-

and-cut algorithms (Naddef & Rinaldi, 2001) provide exact solutions but suffer from high computational complexity at scale, making them impractical for large problems. Heuristic reviews (Cordeau et al., 2002) cover savings algorithms, sweep methods, and graph-based approaches; these perform well for small instances but prove inefficient for larger, real-world problems due to scalability issues. While effective for small instances, these methods struggle with computational scalability and generalization for modern large-scale VRPs.

## 2.2 AI Methods for Large-Scale VRP

AI techniques have significantly advanced VRP solving through Graph Convolutional Networks (Gasse et al., 2019), attention models (Kool et al., 2019), and neural predictors (Accorsi & Vigo, 2021). Key innovations include Neural Large Neighborhood Search (Chen et al., 2020), learning-to-delegate frameworks (Li et al., 2021), TAM's route decomposition (Hou et al., 2023), and PolyNet's diverse solution learning (Hottung et al., 2024). CaDA (Li et al., 2025b) introduces constraint-aware dual-attention for cross-problem VRP solving, achieving state-of-the-art results across 16 VRP variants through constraint prompts and selective attention mechanisms. While demonstrating superior generalization, these methods often require substantial training resources.

## 2.3 Multi-Task VRP Problems

Recent advances in multi-task Vehicle Routing Problems focus on developing integrated architectures capable of handling multiple constraints and variants. Key approaches include joint attention, reinforcement learning, and Mixture-of-Experts models. Joint attention mechanisms capture dependencies between tasks. Falkner & Schmidt-Thieme (2020) uses joint attention for multi-constraint coordination. Reinforcement learning enables dynamic policy adaptation. Delarue et al. (2020) demonstrates its robustness in varying scenarios. Mixture-of-Experts models specialize in different problem variants; Zhou et al. (2024) proposes a multi-task MoE framework. These methods build on foundational work: Shazeer et al. (2017) introduces efficient MoE routing, while Berto et al. (2025b) develops a unified Transformer-based platform. The survey by Wu et al. (2025a) categorizes paradigms and highlights challenges like generalization and comparison difficulties. Evaluations show improved multi-task performance, though expert balancing and training stability remain challenges, pointing to future work in adaptive gating and scalable training.

## 3 Preliminary Work

### 3.1 Markov Decision Process Model for VRP

We model the VRP as a Markov Decision Process (MDP) $(\mathcal{S}, \mathcal{A}, \mathcal{T}, R, \gamma)$ where states include node positions and constraints, actions select customer nodes, and rewards are negative distances. The goal is to maximize expected cumulative reward while respecting capacity and other constraints.

The policy $p_\theta(a_t|s_t)$ parameterized by $\theta$ defines the probability of selecting action $a_t$ in state $s_t$. For the original VRP with $n$ nodes, this policy generates a complete solution sequence $S = [a_1, a_2, \ldots, a_n]$ that minimizes the total cost $\mathcal{C}(S) = \sum_{i=1}^{n-1} d_{a_i, a_{i+1}}$, where $d_{i,j}$ denotes the Euclidean distance between node $i$ and node $j$, subject to constraints like $\sum_{i \in S_k} q_i \leq Q$ for each vehicle route $S_k$. The full policy is expressed as:

$$p_\theta(S|s) = \prod_{t=1}^{n} p_\theta(a_t|s_t), \tag{1}$$

where the sequence length is fixed to $n$. Mathematical formulations are in Appendix B.1.

### 3.2 TAM and MVMoE

Hou et al. (2023)'s TAM decomposes problems into parallel small-scale TSPs via sequence-to-sequence sub-route generation followed by parallel optimization. Zhou et al. (2024)'s MVMoE uses MoE gating, multi-task loss, and REINFORCE optimization for multi-variant VRPs. While efficient for small instances (<200 nodes), it degrades on larger problems due to training data limitations. Its offline training enables real-time applications like dynamic scheduling. Technical details are in Appendix B.2 and B.3.

# 4 METHOD

## 4.1 GENERAL FORMULATION OF PAML

In PAML, we reformulate the VRP solving process as a three-stage policy that decomposes the problem into subproblems, merges them, and solves them in parallel. The overall policy $p_\theta(S^*|s)$ generates a partitioned and merged sequence leading to the best found solution $S^*$ for the initial state $s$, where $\theta$ parameterizes the partitioner. The overall policy is expressed as:

$$p_\theta(S^*|s) = p_\theta(\pi|s) \cdot p(\mathcal{M}|\pi) \cdot p(S^*|\mathcal{M}) \tag{2a}$$

where

$$p_\theta(\pi|s) = \prod_{t=1}^{T} p_\theta(\pi_t|s_t) \tag{2b}$$

$$p(\mathcal{M}|\pi) = \delta(\mathcal{M} = \mu(\zeta(\pi))) \tag{2c}$$

$$p(S^*|\mathcal{M}) = \prod_{j=1}^{l} \delta(S_j^* = \Psi(\mathcal{M}_j)) \tag{2d}$$

In these expressions, $\pi = [\pi_1, \pi_2, \ldots, \pi_T]$ denotes the initial partition sequence with $T$ steps including separators, and $s_t$ is the state at step $t$; $m$ is the number of initial subproblems; $P_k$ obtained by splitting $\pi$ at separators via $\mathcal{P} = \zeta(\pi) = [P_1, P_2, \ldots, P_m]$; $\mathcal{M} = [\mathcal{M}_1, \mathcal{M}_2, \ldots, \mathcal{M}_l]$ is the vector of merged subproblems obtained by applying the merging function $\mu$ to the initial partition $\mathcal{P}$; $l$ is the number of merged subproblems; $S^* = \{S_1^*, S_2^*, \ldots, S_l^*\}$ denotes the set of best found sub-solutions; and $S_j^*$ is the best found sub-solution for $\mathcal{M}_j$ under subproblem solver $\Psi$. Here, $\delta(\cdot)$ denotes the Kronecker delta function, which equals 1 when the condition inside is satisfied and 0 otherwise, representing deterministic operations in the merging and solving stages; $\zeta(\cdot)$ is the splitting function that partitions the sequence at separators; and $\mu(\cdot)$ is the merging function that consolidates the initial subproblems from $\mathcal{P}$ into a vector of merged subproblems.

This formulation evolves the original policy $p_\theta(S^*|s)$ by introducing decomposition: $p_\theta(\pi|s)$ produces a sequence of nodes interspersed with separators, transforming the fixed-length sequence into a variable-length one with delimiters; $p(\mathcal{M}|\pi)$ combines initial subproblems based on geometric heuristics, further adapting the policy to handle grouped subsets; and $p(S^*|\mathcal{M})$ applies a pre-trained solver to each merged subproblem, enabling parallel evaluation.

This formulation enables parallel computation and improves generalization by reducing the effective problem size while preserving global constraints.

## 4.2 DETAILED METHODOLOGY

### 4.2.1 GENERATING INITIAL SUBPROBLEM SEQUENCE

The partitioner generates a sequence $\pi = [\pi_1, \pi_2, \ldots, \pi_T]$ mixing customer nodes and separator tokens (0), where $T \approx n + n/n_{\text{target}}$, $n_{\text{target}}$ denotes the target subproblem size. The policy becomes:

$$p_\theta(\pi|s) = \prod_{t=1}^{T} p_\theta(\pi_t|s_t), \tag{3}$$

with constraint masking:

$$\text{Mask}(a_t) = \begin{cases} 0 & \text{if } \sum_{i \in \mathcal{R}_t} q_i + q_{a_t} \leq Q \\ -\infty & \text{otherwise,} \end{cases} \tag{4}$$

where $\mathcal{R}_t$ denotes the set of nodes currently assigned to the active route at step $t$. Initial subproblems are obtained by splitting at separators: $\mathcal{P} = \zeta(\pi) = [P_1, P_2, \ldots, P_m]$.

### 4.2.2 SUBPROBLEM MERGING

The merging stage consolidates initial subproblems $\mathcal{P}$ into $\mathcal{M} = [\mathcal{M}_1, \mathcal{M}_2, \ldots, \mathcal{M}_l]$ using geometric heuristics. Subproblems are sorted by polar angle $\alpha_k = \arctan 2(\bar{y}_k - y_{\text{depot}}, \bar{x}_k - x_{\text{depot}})$ relative to depot, then merged using either fixed-number or target node count strategies. Here $(\bar{x}_k, \bar{y}_k)$ denotes the centroid of the $k$-th initial subproblem. After merging, the consolidated subproblems $\mathcal{M}$ are ready for parallel solving, where each $\mathcal{M}_j$ will be processed by a pre-trained solver to obtain the optimal sub-solution $S_j^*$.

### 4.2.3 PARALLEL SOLVING OF MERGED SUBPROBLEMS

The merged subproblems $\mathcal{M}$ are solved in parallel using a pre-trained multi-task solver. For each $\mathcal{M}_j$ with size $n_j \approx n_{\text{target}}$, pad to uniform size if needed and compute the sub-solution $S_j^*$ that minimizes the local cost $\mathcal{C}(S_j^*)$ while satisfying subset constraints. The policy for solving adapts to subsets:

$$p(S^*|\mathcal{M}) = \prod_{j=1}^{l} \delta(S_j^* = \Psi(\mathcal{M}_j)), \tag{5}$$

where $\Psi$ is a fixed pre-trained solver that deterministically computes the best found solution for each subproblem $\mathcal{M}_j$. The overall solution is the concatenation calculated as:

$$S^* = \bigoplus_{j=1}^{l} S_j^*, \tag{6}$$

with total cost:

$$\mathcal{C}_{\text{total}}^* = \sum_{j=1}^{l} \mathcal{C}(S_j^*). \tag{7}$$

### 4.2.4 INFERENCE PIPELINE

The optimized inference flow (Fig. 2) is: 1) **Partition**: Generate initial sequence $\pi$ via greedy decoding with constraint masking. 2) **Merge**: Merge into $\mathcal{M}$ using target-size strategy ($n_{\text{target}}$ calibrated per problem scale in Table 1). 3) **Pad**: Uniform padding for batch processing. 4) **Solve**: Solve all subproblems in parallel to obtain $S_j^*$. 5) **Combine**: Concatenate sub-solutions $S^* = \bigoplus S_i^*$.

### 4.2.5 END-TO-END TRAINING OF THE PARTITIONER

The partitioner parameters $\theta$ are optimized using REINFORCE with greedy baseline $\pi^{\text{BL}}$:

$$\nabla J(\theta) = \mathbb{E}_{\pi \sim p_\theta} \left[ (R(\pi) - R(\pi^{\text{BL}})) \nabla \log p_\theta(\pi) \right], \tag{8a}$$

where

$$R(\pi) = -\sum_{j=1}^{l} \mathcal{C}(S_j^*) + \lambda \sum_{j=1}^{l} \left( \beta_{\text{cap}} \mathcal{V}_{\text{cap}}(\mathcal{M}_j) + \beta_{\text{tw}} \mathcal{V}_{\text{tw}}(\mathcal{M}_j) + \beta_{\text{route}} \mathcal{V}_{\text{route}}(\mathcal{M}_j) \right), \tag{8b}$$

with violation terms $\mathcal{V}_{\text{cap}}(\mathcal{M}_j) = \max(0, \sum_{i \in \mathcal{M}_j} q_i - Q)$, $\mathcal{V}_{\text{tw}}(\mathcal{M}_j) = \sum_{i \in \mathcal{M}_j} \max(0, t_i - l_i)$, and $\mathcal{V}_{\text{route}}(\mathcal{M}_j) = \max(0, \text{Length}(\mathcal{M}_j) - L_{\max})$ for capacity, time window, and route length constraints respectively. The total loss incorporates MoE load balancing:

$$\mathcal{L}_{\text{total}} = \mathcal{L}_{\text{RL}} + \lambda_m \mathcal{L}_{\text{MoE}}. \tag{9}$$

Here $\text{Length}(\cdot)$ measures the total route length within a subproblem in the same units as $\mathcal{C}(\cdot)$.

## 5 EXPERIMENTS

### 5.1 EXPERIMENTAL SETUP

#### 5.1.1 DATA GENERATION

The data generation method is designed to cover vehicle routing problem (VRP) variants of varying scales and complexities to ensure the model's generalization capability. Taking the classic Capacitated Vehicle Routing Problem (CVRP) as an example, the data generation process constructs problem instances using a randomized approach. The geographical locations of nodes are generated through uniform random sampling. For each problem instance, the depot coordinates are randomly generated within a two-dimensional unit plane ($[0, 1] \times [0, 1]$), while customer node coordinates also follow a uniform distribution. For problems of different scales (e.g., 20, 50, 100, 200 nodes), the generator dynamically adjusts the baseline capacity value. For instance, the baseline capacity is set to 30 for 20-node problems, 40 for 50-node problems, 50 for 100-node problems, and up to 200 for 2000-node problems. As the problem size increases, the baseline capacity is scaled accordingly

324 to ensure an appropriate level of difficulty. Customer demands are normalized by first generating
325 random integers between 1 and 10 and then dividing them by the baseline capacity, thereby trans-
326 forming demand constraints into continuous values within the $[0, 1]$ range. For other VRP variants
327 (e.g., OVRP, VRPB), the generator extends the randomization logic of CVRP with problem-specific
328 constraints. For example, in the Vehicle Routing Problem with Backhauls (VRPB), 20% of cus-
329 tomer nodes are randomly selected as backhaul nodes, with their demand values set as negative to
330 distinguish them from linehaul deliveries. Additionally, the route length limit is uniformly set to 3.0
331 as a global constraint on vehicle travel distance.

### 5.1.2 BASELINE METHODS AND EVALUATION METRICS

We compare our proposed method with several baseline approaches: 1) **MVMoE**: Directly solv-
ing the full original VRP instance using the MVMoE solver. 2) **OR-Tools**: Solving the complete
original VRP instance using Google's OR-Tools. 3) **PAML+MVMoE**: The original VRP instance
is decomposed using PAML and solved using MVMoE solver. 4) **PAML+OR-Tools**: The origi-
nal VRP instance is decomposed using PAML and solved using OR-Tools. The evaluation metrics
include solution quality (total route length, with smaller values being better) and computational effi-
ciency (solving time in seconds). We conduct experiments on both generated datasets and real-world
instances from CVRPLIB, with problem sizes ranging from 50 to 2000 nodes.

### 5.2 MAIN RESULTS

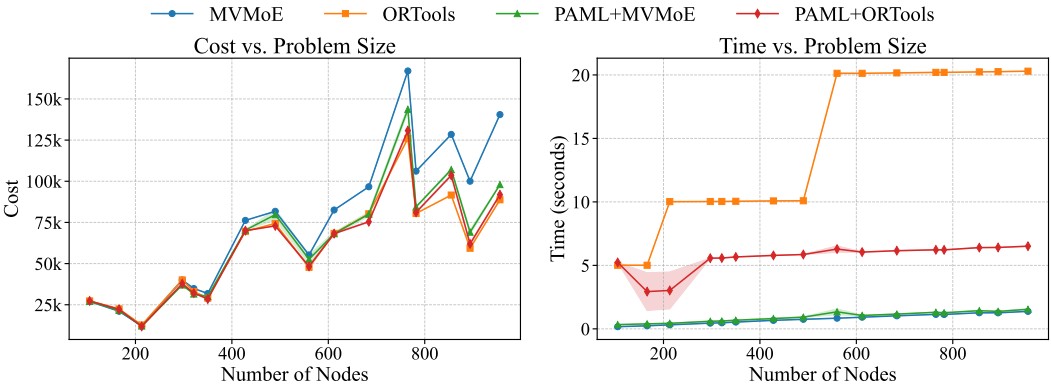

Figure 4: Solving cost (smaller is better) and solving time (smaller is better) on CVRPs of different
sizes (100 to 1000-nodes) from CVRPLIB datasets using four different methods.

### 5.2.1 PERFORMANCE ON GENERATED DATA

We tested the solution results on five VRP variants (CVRP, OVRP, VRPB, VRPL, and VRPTW)
with problem sizes ranging from 50 to 2000. Our experimental results demonstrate that the PAML
method exhibits significant advantages in solving VRPs of varying scales, particularly when com-
bined with the MVMoE Solver: 1) For 400-node and smaller problems: PAML+MVMoE achieves
optimal performance, reducing route length by 3.40% compared to MVMoE Solver while maintain-
ing computation time at only 10% of OR-Tools. 2) For 500-node problems: Solution quality matches
OR-Tools while reducing route length by 8.67% compared to MVMoE Solver. 3) For 1000-node
problems: Route length reduced by 20.66% compared to MVMoE Solver, with computation time
under 1.58 seconds (92.23% faster than OR-Tools). 4) For 2000-node problems: Route length re-
duced by 48.71% compared to MVMoE Solver, with just 3.62 seconds computation time (91.20%
faster than OR-Tools).

Table 3 that presents detailed data for VRPs for all scales , and Figure 6 that shows the performance
of all methods on different VRP variants are included in Appendix D.2. The statistical analysis in
Figure 8 confirms the significance of these improvements, showing consistent performance gains
across problem scales with $p < 0.05$ significance levels. Complete instance-level results and route
visualizations are available in Appendix D.3.

### 5.2.2 PERFORMANCE ON REAL-WORLD CVRPS

To validate our method's effectiveness in real-world scenarios with irregular customer distributions, we tested PAML on the CVRPLIB dataset. The results, shown in Figure 4, align with our findings on generated data: 1) For medium-scale instances (200-400 nodes): PAML+MVMoE achieves the shortest routing distances while maintaining solving times under 0.9 seconds, significantly outperforming OR-Tools' average time of 5.5 seconds. 2) For larger instances (100-1000 nodes): PAML+MVMoE shows a 15.75% reduction in routing distance compared to MV-MoE Solver (63261.76 vs 75091.50), with only a modest 6.43% increase compared to OR-Tools (63261.76 vs 59438.02). Notably, the computational efficiency remains substantial, with average solving time being merely 6.55% of OR-Tools' (0.95s vs 14.50s).

Detailed results are provided in Appendix D.3. Table 4 provides detailed instance-level results, Figure 7 illustrates the distribution of costs and computation times across methods, and Figure 8 further validates the statistical significance of these improvements across different problem scales. Route visualizations are available in Appendix D.4.

## 5.3 ABLATION STUDIES

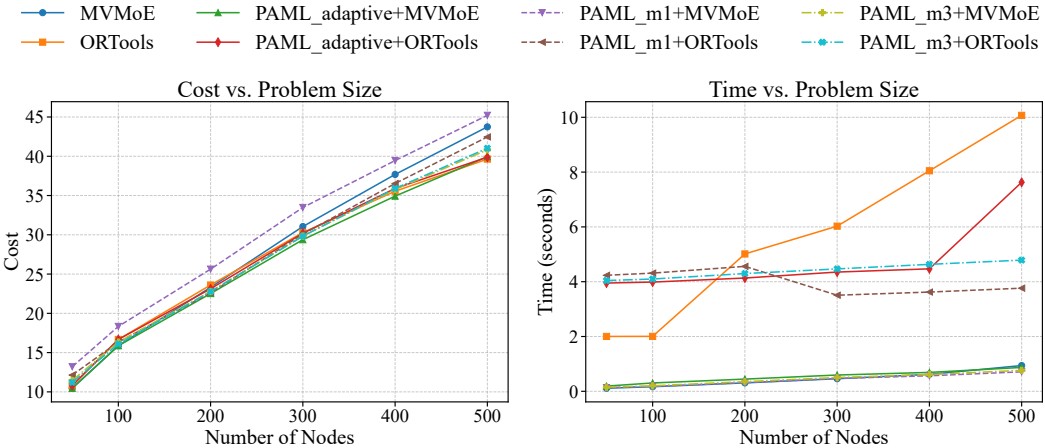

Figure 5: Solving cost (left, smaller is better) and solving time (right, smaller is better) on VRPs of different sizes (200 to 500-nodes) using all methods.

As shown in Figure 5, in order to assess the effectiveness of different components in our framework, we conducted ablation studies focusing on the merging strategies. We compared three core approaches: 1) **No Merging (PAML_m1)**: Baseline approach solving the partitioner's raw sub-routes directly. 2) **Fixed Number Merging (PAML_m3)**: Merging a fixed quantity of sorted primitive sub-routes. 3) **Adaptive Merging (PAML_adaptive)**: Merging based on a target node count.

Results for 1000-node VRPs when combined with MVMoE solver show: 1) MVMoE: Average path length 79.69. 2) No Merging: Path length 80.48 (-0.98% improvement). 3) Fixed Subproblem Number Merging: Path length 66.01 (17.09% improvement). 4) Adaptive Subproblem Size Merging: Path length 63.23 (20.66% improvement). These results demonstrate that: 1) Merging is essential for solution quality. 2) Fixed Subproblem Number Merging significantly improves over no merging. 3) Adaptive Subproblem Size Merging consistently performs best across all problem sizes. Detailed performance data is available in Appendix D.2.

## 5.4 ANALYSIS OF BEST SUBPROBLEM MERGE SIZE

Building on the ablation study results, we further analyzed the best target size parameter for Adaptive Merging. While our solver model has a native training size of 50 nodes, experiments revealed that uniformly decomposing large-scale VRPs into 50-node subproblems leads to global information loss.

Table 1: Best Subproblem Merge Size Information for Different VRP Variants and Problem Scales

| Variant | Problem Size | Best SP Size | Cost | Avg SPs | Avg Time |
|---------|--------------|--------------|------|---------|----------|
| CVRP | 200
500
1000
2000 | 100
125
150
300 | 23.22
41.96
65.82
128.95 | 2.00
4.00
6.36
6.00 | 0.27
0.64
1.26
2.58 |
| OVRP | 200
500
1000
2000 | 100
125
150
300 | 16.26
30.67
51.40
100.37 | 2.00
4.00
6.04
6.00 | 0.27
0.65
1.27
2.68 |
| VRPB | 200
500
1000
2000 | 100
125
200
300 | 18.44
34.07
59.42
119.76 | 2.00
4.04
5.02
6.30 | 0.29
0.68
1.37
2.78 |
| VRPL | 200
500
1000
2000 | 100
125
150
300 | 23.82
42.59
67.35
120.84 | 2.00
4.00
6.24
6.00 | 0.29
0.70
1.37
2.95 |
| VRPTW | 200
500
1000
2000 | 20
50
50
100 | 107.54
284.06
572.59
1184.47 | 9.74
9.00
18.00
17.18 | 0.35
0.86
1.72
3.57 |

As shown in Table 1, our systematic analysis shows that the best merging size exhibits a sublinear relationship with the problem scale. For example, in CVRP with N=2,000, the best subproblem merging target (Best SP Size, SP here means subproblem) is 300 (approximately 15% of the original problem size). This finding balances the trade-off between retaining global information and maintaining solver efficiency.

## 6  CONCLUSION

This paper introduces an innovative framework, named PAML, which employs a novel divide-and-conquer strategy to efficiently solve large-scale, multi-variant Vehicle Routing Problems (VRPs). The core innovation of this framework lies in the organic integration of two key technologies: an intelligent, Mixture-of-Experts (MoE) based implicit partitioner trainable via end-to-end reinforcement learning, and a dynamic subproblem merging mechanism based on polar angle clustering. The partitioner dispenses with manually designed rules to adaptively generate high-quality division schemes, while the merging mechanism intelligently regulates subproblem sizes, striking a delicate balance between preserving global information and maintaining solver efficiency. This approach effectively overcomes the core challenges of computational complexity, generalization, and multi-task optimization that plague existing methods for complex VRPs.

Comprehensive experimental results clearly demonstrate the superior performance and practical value of the PAML framework, highlighted by three key advantages: 1) Real-time Computational Efficiency: Compared to industry-standard optimizers like OR-Tools, PAML reduces computation time to just one-tenth while maintaining high-quality solutions, showcasing its immense potential in dynamic scenarios requiring rapid decision-making. 2) Strong Generalization to Large-Scale VRPs: While the performance of the baseline MVMoE solver degrades significantly on instances with thousands of nodes, the PAML framework exhibits robust generalization. It achieves up to a 48.71% reduction in path cost compared to solving with MVMoE alone, proving its robustness as problem scale increases. 3) Superior Multi-Task Optimization: Across various VRP variants, PAML consistently delivers performance improvements. This is attributable to its MoE-based architecture, which enables flexible adaptation to the unique constraints of different tasks, validating its effectiveness as a universal VRP solving framework.

In summary, the PAML framework not only provides an efficient and scalable solution for large-scale VRPs but, more importantly, establishes a new paradigm of synergistic design between neural partitioning and expert models. This work offers valuable insights and reusable techniques for the intelligent solving of other combinatorial optimization problems.

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
