APPENDIX

APPENDIX CONTENTS

## A LLM USAGE STATEMENT

In the process of preparing this manuscript, large language models (LLMs) were utilized as a general-purpose assistive tool. Specifically, the LLMs were employed to aid in the writing and polishing of the manuscript, as well as to assist with code-related tasks. It is important to note that all research ideas, theoretical work, and findings, as well as the design and execution of model training and experiments, were solely developed and conducted by the authors. The authors take full responsibility for the content of this paper, and adhere to the standards of academic integrity.

## B TECHNICAL PRELIMINARIES

### B.1 VRP MATHEMATICAL FORMULATION

Given a depot node $v_0$ and $n$ customer nodes $V = \{v_1, v_2, ..., v_n\}$, each with demand $q_i$, vehicle capacity $Q$, and distance matrix $D = [d_{ij}]$, the objective function is the following:

$$\min \sum_{k=1}^{K} \sum_{i=0}^{n} \sum_{j=0}^{n} d_{ij} x_{ijk} \tag{10}$$

Where $K$ is total number of vehicles; $d_{ij}$ is distance or cost from node $i$ to $j$; $x_{ijk}$ is binary variable indicating whether vehicle $k$ travels from $i$ to $j$. Each customer is served exactly once:

$$\sum_{k=1}^{K} \sum_{i=0}^{n} x_{ijk} = 1, \quad \forall j = 1, 2, \ldots, n \tag{11}$$

Variant-Specific Constraints:

#### B.1.1 CVRP (CAPACITATED VRP)

Vehicle capacity constraint: The total demand served by each vehicle must not exceed its capacity $Q$.

#### B.1.2 OVRP (OPEN VRP)

No return to depot: Vehicles do not need to return to the depot after serving customers.

### B.1.3 VRPB (VRP with Backhauls)

Customers are divided into linehaul (delivery, $q_i > 0$) and backhaul (pickup, $q_i < 0$). Linehaul before backhaul in each route: For each vehicle, if a backhaul node is visited, all linehaul nodes in the route must precede it.

### B.1.4 VRPL (VRP with Route Length Limit)

Route length constraint: The total length of each route must not exceed the maximum limit $L_{\max}$.

### B.1.5 VRPTW (VRP with Time Windows)

Each customer has a time window $[e_i, l_i]$, and the arrival time $t_i$ must satisfy $e_i \le t_i \le l_i$. Additionally, time propagation constraints ensure that the arrival time at each subsequent customer accounts for travel time and service time from the previous one.

## B.2 TAM Technical Details

Key innovations include: 1) The MDP action space is redefined to reduce the impact of node order. 2) A novel reward function uses the best sub-route length. 3) Two-stage training incorporates TSP solver pre-training.

**Sub-route Sequence Generation** By redefining the action space of the Markov Decision Process (MDP) as sub-routes rather than individual nodes, the impact of node order on model generalization is mitigated. The core formula is:

$$p_\theta(r \mid s) = \prod_{t=1}^{l} p_\theta(r_t \mid s_t) \tag{12}$$

Where $r$ is the set of sub-route sequences; $r_t$ is the $t^{\text{th}}$ sub-route, representing a set of nodes served by the same vehicle; $s_t$ is the state at step $t$, containing information about the currently assigned sub-routes and unassigned nodes; and $\theta$ is the parameters of the policy network, used to model the probability distribution of sub-route selection.

**Novel Reward Function** The best length of a sub-route is used as the reward to ensure it is independent of node order:

$$R_i = -\min_{\varphi \in \Phi} \text{dist}\big(\varphi(r_i)\big) \tag{13}$$

Where $R_i \in \mathbb{R}$ is the reward for sub-route $r_i$, $\Phi$ is the set of all possible node permutations, $\varphi(r_i)$ is a permuted sequence of nodes in $r_i$, and $\text{dist} : \mathcal{P} \to \mathbb{R}^+$ computes route length.

A global mask function is also introduced to enforce constraints such as the maximum number of vehicles:

$$u_0 = \begin{cases} \tanh\left(\frac{q^\top k_0}{\sqrt{d_{hid}}}\right), & \text{if } \sum_{i \in \Omega_u} q_i \le Q \cdot (l_m - l_u) \\ -\infty, & \text{otherwise} \end{cases} \tag{14}$$

Where $u_0$ is the depot selection probability, $q$ and $k_0$ are the attention vectors, $d_{hid}$ is the hidden dimension, $\Omega_u$ are the unassigned nodes, $q_i$ are the demands, $Q$ is vehicle capacity, $l_m$ is the max number of vehicles, and $l_u$ is the number of used vehicles.

**Two-stage Training and Padding Technique** The reinforcement learning process is accelerated by pre-training a TSP solver and employing a padding technique, while parallel computing is leveraged to optimize sub-route solving efficiency.

## B.3 MVMoE Technical Details

**MoE Gating Mechanism** Input features $h$ pass through gating network $G$ to compute expert weights $g$:

$$g = \text{TopKSoftmax}(G(h)) \tag{15}$$

MoE output is the weighted sum of experts:

$$\text{MoE}(h) = \sum_{i=1}^{E} g_i \cdot \text{Expert}_i(h) \tag{16}$$

Where $h$ is the input features (e.g., node embeddings, task embeddings), $G(\cdot)$ is the gating network (typically a fully connected layer), $g$ is the expert weight vector (activation levels), $E$ is the total number of experts, and $\text{Expert}_i(\cdot)$ is the $i^{\text{th}}$ expert network.

**Multi-Task Loss Function** For $num_{tasks}$ tasks with losses $L_t$, the total loss is:

$$L_{\text{total}} = \sum_{t=1}^{num_{tasks}} w_t L_t \tag{17}$$

Where $num_{tasks}$ is the total number of tasks (e.g., CVRP, VRPTW), $L_t$ is the loss function for task $t$, and $w_t$ is the weight for task $t$'s loss.

**Reinforcement Learning Objective** Using REINFORCE to optimize parameters $\theta$ with reward $R(S) = -\mathcal{C}(S)$:

$$\nabla_\theta J(\theta) = \mathbb{E}_{S \sim p_\theta} \left[ (R(S) - b) \nabla_\theta \log p_\theta(S) \right] \tag{18}$$

Where $\theta$ is the model parameters (partitioner or solver), $S$ is the sampled node sequence or solution, $p_\theta(S)$ is the probability of generating sequence $S$ under policy $p_\theta$, $R(S)$ is reward ($R(S) = -\mathcal{C}(S)$, where $\mathcal{C}(S)$ is total cost), $b$ is the baseline (reduces variance).

## C  MODEL ARCHITECTURE AND IMPLEMENTATION DETAILS

### C.1  DETAILED IMPLEMENTATION OF THE PARTITION FRAMEWORK

**Detailed Architecture of Implicit Partitioner**

The framework's multi-task processing relies on three components: constraint unification encoding, variant-aware attention, and hierarchical feature learning.

The constraint unification encoding maps various VRP constraints (capacity, time windows, back-hauls) into unified numerical representations. The variant-aware attention mechanism dynamically adjusts attention distributions according to problem types. Hierarchical feature learning first learns general spatial features through multi-layer abstraction, then learns variant-specific constraint patterns.

The framework combines parameter sharing with task-specific adaptation. Bottom layers share parameters across VRP variants, learning universal spatial relationships and optimization principles. Middle layers introduce variant-specific modules that selectively activate feature channels through gating mechanisms. The top layer combines general and variant-specific knowledge to generate constraint-compliant partition sequences.

**Input and Output Details** The partitioner input includes depot location, customer nodes, demands, service times, time windows, and constraints (capacity, route limits), encoded as tensors. Output is a node visitation sequence with customer indices and separators (0).

**Sequence Generation Process** The model performs sequence construction, outputting probability distributions for the next node (or separator 0) based on current state. Actions are selected via sampling or greedy strategy.

**Encoder** Uses hierarchical feature extraction: spatial encoding layers learn geometric relationships, constraint encoding layers process variant constraints, and variant identification layers generate problem-specific representations.

**Decoder** Uses attention mechanisms to fuse encoding features, historical decisions, and constraint states, generating constraint-compliant probability distributions.

## C.2 HEURISTIC METHODS FOR SUBPROBLEM MERGING

The partitioner may generate numerous small subproblems that are inefficient to solve directly. A merging step consolidates adjacent or similar initial subproblems into reasonably-sized subproblems.

**Basic Flow** Extract subproblems separated by 0 from partitioner output. Merge using either centroid distance (closest pairs) or polar angle (depot-relative sorting) until target size/number is reached.

*Centroid Distance Merging* Calculate Euclidean distances between subproblem centroids, merge the closest pairs until the target number or size is met.

*Polar Angle Merging* Using the depot as the origin, calculate the polar angle of each subproblem's centroid, then merge adjacent subproblems after sorting by polar angle until the target number or size is met.

**Heuristic Method 1: Merging by Fixed Number** This method aims to merge a fixed number of initial subproblems, sorted by their polar angles, into a larger subproblem.

*Parameter* Let MergeNum $\in \mathbb{N}$ denote how many consecutive, polar-angle-sorted primitive subproblems constitute each finally merged subproblem.

*Process* The system obtains a list of initial subproblems that have undergone centroid calculation and polar angle sorting. It then sequentially takes MergeNum initial subproblems from this list, combining all customer nodes to form a new merged subproblem. This is repeated until all initial subproblems are assigned. If fewer than MergeNum subproblems remain, they are merged into the last subproblem.

*Termination Condition* The merging process terminates when all initial subproblems have been processed. The number of subproblems generated is approximately $\frac{n}{MergeNum}$, where $n$ is the number of initial subproblems.

*Characteristics* This method is simple, intuitive, and allows easy control over the number of final subproblems. However, it does not directly consider the node scale of the merged subproblems, potentially leading to uneven sizes in the final subproblems.

**Heuristic Method 2: Merging by Target Node Count** This method creates subproblems with a node count close to a predetermined target, better controlling the scale of instances fed to the subproblem solver.

*Parameters* $n_{\text{target}}$, which specifies the desired number of nodes for a merged subproblem, typically triggered by a special value.

*Process* Initial subproblems are subjected to centroid calculation and polar angle sorting. The system constructs the first merged subproblem by iteratively adding initial subproblems from the sorted list, checking the total node count after each addition.

*Merge Termination/Switching Conditions* If the node count reaches or slightly exceeds $n_{\text{target}}$, the current group forms a final subproblem. If adding the next subproblem would significantly exceed $n_{\text{target}}$, the current group is stopped. This process is repeated until all initial subproblems are assigned.

*Characteristics* This method allows finer control over the scale of final subproblems, maximizing the performance of the subproblem solver and adjusting the granularity based on the solver's optimal scale.

**Depot Node Handling** When constructing subproblem $\mathcal{M}_j$, explicitly include the depot node $v_0$:

$$\mathcal{M}_j = \{v_0\} \cup \mu(\mathcal{P})_j \tag{19}$$

The solver $p_\phi(S_j^* | \mathcal{M}_j)$ processes the complete VRP instance where each subproblem $\mathcal{M}_j$ includes the depot node $v_0$ and the customer nodes from the j-th merged subproblem.

## C.3 TRAINING ALGORITHM

---

**Algorithm 1** Implicit Partitioner Training Algorithm

---

**Input**: Maximum epochs $E_{\text{ep}}$, instances per epoch $N$, batch size $B$, learning rate $\eta$, MoE coefficient $\lambda_m$, accumulation steps $S$, pre-trained solver $\Psi$;
**Output**: Trained partitioner $\Theta$.

1:  Initialize parameters $\theta$ of $\Theta$
2:  Initialize optimizer $O$ (e.g., Adam)
3:  **for** epoch $e = 1$ to $E_{\text{ep}}$ **do**
4:      Set $\Theta$ to training mode
5:      $O$.zero_grad(), $k \leftarrow 0$
6:      **for** batch $b = 1$ to $\lceil N/B \rceil$ **do**
7:          Sample batch $\{I_j\}_{j=1}^{B}$ from training data
8:          Initialize $\mathcal{L}_b$
9:          **for** each instance $I_j$ in batch **do**
10:             **Sampling Rollout:**
11:             Generate $\pi_j$ using $\Theta(\theta)$ via sampling
12:             Record $\log p_\theta(\pi_j | I_j) = \sum_t \log p_\theta(a_{j,t} | s_{j,t})$
13:             **Greedy Baseline:**
14:             Generate $\pi_j^{\text{BL}}$ via greedy selection
15:             **Reward Computation:**
16:             Compute $R(\pi_j)$ and $R(\pi_j^{\text{BL}})$ using $\Psi$
17:             Calculate advantage $A_j = R(\pi_j) - R(\pi_j^{\text{BL}})$
18:             $\ell_j \leftarrow -A_j \cdot \log p_\theta(\pi_j | I_j)$
19:             Add $\ell_j$ to $\mathcal{L}_b$
20:         **end for**
21:         $\mathcal{L}_r \leftarrow \text{mean}(\mathcal{L}_b)$
22:         $\mathcal{L}_t \leftarrow \mathcal{L}_r$
23:         **if** $\Theta$ contains MoE layers **then**
24:             Get $\mathcal{L}_m$ from expert balancing
25:             $\mathcal{L}_t \leftarrow \mathcal{L}_t + \lambda_m \mathcal{L}_m$
26:         **end if**
27:         $\mathcal{L}_s \leftarrow \mathcal{L}_t / S$
28:         $\mathcal{L}_s$.backward()
29:         $k \leftarrow k + 1$
30:         **if** $k \geq S$ **then**
31:             **Optional:** Gradient clipping
32:             $O$.step()
33:             $O$.zero_grad()
34:             $k \leftarrow 0$
35:         **end if**
36:     **end for**
37:     **if** remaining gradients ($k > 0$) **then**
38:         **Optional:** Gradient clipping
39:         $O$.step()
40:         $O$.zero_grad()
41:     **end if**
42:     Adjust $\eta$ per schedule
43:     **if** $e$ meets validation interval OR $e = E_{\text{ep}}$ **then**
44:         Evaluate on validation set
45:     **end if**
46:     **if** $e$ meets saving interval OR $e = E_{\text{ep}}$ **then**
47:         Save $\theta$
48:     **end if**
49: **end for**

---

**Variable Definitions:** $E_{\text{ep}}$: maximum training epochs; $N$: instances per epoch; $B$: training batch size; $\eta$: learning rate; $\lambda_m$: MoE loss coefficient; $S$: gradient accumulation steps; $\Psi$: fixed pre-trained subproblem solver; $\Theta$: partitioner policy network; $\theta$: model parameters; $O$: optimizer; $k$: accumulation counter; $I_j$: VRP instance $j$; $\mathcal{L}_b$: batch instance losses; $\pi_j$: sampled partition for instance $j$; $a_{j,t}$: action at step $t$ for instance $j$; $s_{j,t}$: state at step $t$ for instance $j$; $\pi_j^{\text{BL}}$: greedy

baseline partition; $A_j$: advantage for instance $j$; $\ell_j$: RL loss for instance $j$; $\mathcal{L}_r$: reinforcement learning loss; $\mathcal{L}_t$: total batch loss; $\mathcal{L}_m$: MoE auxiliary loss; $\mathcal{L}_s$: scaled loss for backpropagation.

**Algorithm Description:** The algorithm outlines the training procedure for the implicit partitioner using REINFORCE with a Greedy Rollout Baseline to optimize the partitioner policy network $\Theta$. Training begins by initializing the partitioner network $\Theta$ and optimizer $O$. For each epoch $e$, the training data is processed in batches of size $B$. Within each batch, the algorithm processes individual VRP instances $I_j$. For each instance, the algorithm generates two partition schemes: a sampled scheme $\pi_j$ using stochastic sampling from $\Theta(\theta)$, and a greedy baseline $\pi_j^g$ using deterministic greedy selection. The log-probabilities of the sampled actions are recorded as $\log p_\theta(a_{j,t}|s_{j,t})$. Both partition schemes are evaluated using the fixed pre-trained solver $\Psi$, yielding rewards $R(\pi_j)$ and $R(\pi_j^g)$. The advantage $A_j = R(\pi_j) - R(\pi_j^g)$ measures the relative performance of the sampled scheme. The REINFORCE loss for instance $j$ is computed as $\ell_j = -A_j \cdot \log p_\theta(\pi_j|I_j)$. The batch loss $\mathcal{L}_r$ averages individual losses $\ell_j$. For MoE architectures, an auxiliary loss $\mathcal{L}_m$ promotes expert load balancing, yielding total loss $\mathcal{L}_t = \mathcal{L}_r + \lambda_m \mathcal{L}_m$. Gradient accumulation over $S$ steps enables efficient large-batch training. The optimizer $O$ updates parameters $\theta$ based on accumulated gradients. Learning rate $\eta$ is adjusted according to a schedule, and model validation occurs at specified intervals. This training procedure steers the partitioner $\Theta$ toward generating partition schemes that enable the solver $\Psi$ to achieve lower solution costs for VRP instances.

## D  EXPERIMENTAL SETUP AND DETAILED RESULTS

### D.1  HARDWARE CONFIGURATION

Table 2: Experimental Hardware Configuration

| Component | Specification |
|---|---|
| **GPU (Experimental)** | |
| Model | NVIDIA GeForce RTX 4090 |
| Quantity Used | 1 |
| Memory | 24 GB |
| CUDA Version | 12.7 |
| **CPU** | |
| Model | AMD EPYC 9654 |
| Cores | 96 physical, 192 logical |
| **Memory** | |
| System RAM | 755 GB DDR5 |
| **Storage** | |
| Capacity | 7.0 TB NVMe SSD |
| Available | 4.8 TB |
| **Software** | |
| OS | Ubuntu 24.04 LTS |
| Kernel | Linux 6.8.0-31-generic |

All experiments were conducted on the hardware configuration specified in Table 2. The GPU acceleration was utilized for neural network training and inference, while the high-core-count CPU facilitated parallel processing of optimization algorithms.

## D.2 PERFORMANCE COMPARISON FOR DIFFERENT VRP SCALES

Table 3: Performance Comparison of Different Methods for Solving VRPs of Various Sizes

| Size | Method | Cost | C_Imp | Time | T_Imp |
|---|---|---|---|---|---|
| 50 | OR-Tools | $10.80 \pm 1.47$ | -3.21% | $2.00 \pm 0.00$ | - |
| | MVMoE | $10.46 \pm 1.39$ | - | $0.11 \pm 0.09$ | 94.42% |
| | PAML_adaptive+OR-Tools | $10.80 \pm 1.47$ | -3.20% | $3.95 \pm 0.10$ | -97.31% |
| | PAML_m1+OR-Tools | $12.14 \pm 1.35$ | -16.08% | $4.23 \pm 0.13$ | -111.45% |
| | PAML_m3+OR-Tools | $11.21 \pm 1.44$ | -7.17% | $4.04 \pm 0.14$ | -101.92% |
| | PAML_adaptive+MVMoE | $10.46 \pm 1.39$ | - | $0.19 \pm 0.10$ | 90.49% |
| | PAML_m1+MVMoE | $13.20 \pm 1.74$ | -26.19% | $0.14 \pm 0.10$ | 93.22% |
| | PAML_m3+MVMoE | $11.45 \pm 1.39$ | -9.47% | $0.15 \pm 0.10$ | 92.46% |
| 100 | OR-Tools | $16.67 \pm 1.79$ | -5.04% | $2.00 \pm 0.00$ | - |
| | MVMoE | $15.87 \pm 1.82$ | - | $0.17 \pm 0.01$ | 91.66% |
| | PAML_adaptive+OR-Tools | $16.67 \pm 1.79$ | -5.04% | $3.99 \pm 0.00$ | -98.82% |
| | PAML_m1+OR-Tools | $16.34 \pm 1.67$ | -2.91% | $4.31 \pm 0.07$ | -115.11% |
| | PAML_m3+OR-Tools | $16.10 \pm 1.81$ | -1.42% | $4.10 \pm 0.07$ | -104.39% |
| | PAML_adaptive+MVMoE | $15.87 \pm 1.82$ | - | $0.30 \pm 0.01$ | 84.83% |
| | PAML_m1+MVMoE | $18.33 \pm 2.17$ | -15.49% | $0.18 \pm 0.01$ | 91.11% |
| | PAML_m3+MVMoE | $16.34 \pm 1.80$ | -2.94% | $0.20 \pm 0.01$ | 89.96% |
| 200 | OR-Tools | $23.61 \pm 2.31$ | -1.27% | $5.01 \pm 0.00$ | - |
| | MVMoE | $23.31 \pm 2.04$ | - | $0.31 \pm 0.00$ | 93.87% |
| | PAML_adaptive+OR-Tools | $23.13 \pm 2.30$ | 0.79% | $4.13 \pm 0.03$ | 17.54% |
| | PAML_m1+OR-Tools | $22.61 \pm 2.17$ | 3.00% | $4.56 \pm 0.05$ | 9.05% |
| | PAML_m3+OR-Tools | $22.77 \pm 2.26$ | 2.34% | $4.30 \pm 0.08$ | 14.28% |
| | PAML_adaptive+MVMoE | $22.52 \pm 2.05$ | 3.40% | $0.45 \pm 0.01$ | 91.11% |
| | PAML_m1+MVMoE | $25.63 \pm 2.16$ | -9.94% | $0.31 \pm 0.01$ | 93.75% |
| | PAML_m3+MVMoE | $22.78 \pm 2.12$ | 2.26% | $0.35 \pm 0.01$ | 92.98% |
| 300 | OR-Tools | $30.29 \pm 2.92$ | 2.46% | $6.03 \pm 0.00$ | - |
| | MVMoE | $31.05 \pm 2.79$ | - | $0.46 \pm 0.01$ | 92.37% |
| | PAML_adaptive+OR-Tools | $30.19 \pm 2.96$ | 2.77% | $4.35 \pm 0.15$ | 27.79% |
| | PAML_m1+OR-Tools | $30.08 \pm 3.34$ | 3.12% | $3.51 \pm 0.11$ | 41.81% |
| | PAML_m3+OR-Tools | $29.83 \pm 3.23$ | 3.94% | $4.47 \pm 0.12$ | 25.88% |
| | PAML_adaptive+MVMoE | $29.37 \pm 2.88$ | 5.40% | $0.59 \pm 0.10$ | 90.14% |
| | PAML_m1+MVMoE | $33.48 \pm 3.44$ | -7.81% | $0.47 \pm 0.10$ | 92.23% |
| | PAML_m3+MVMoE | $29.84 \pm 3.08$ | 3.90% | $0.50 \pm 0.10$ | 91.64% |
| 400 | OR-Tools | $35.54 \pm 3.12$ | 5.68% | $8.05 \pm 0.01$ | - |
| | MVMoE | $37.68 \pm 3.14$ | - | $0.62 \pm 0.00$ | 92.32% |
| | PAML_adaptive+OR-Tools | $35.90 \pm 3.63$ | 4.73% | $4.47 \pm 0.07$ | 44.47% |
| | PAML_m1+OR-Tools | $36.53 \pm 3.85$ | 3.06% | $3.62 \pm 0.03$ | 54.99% |
| | PAML_m3+OR-Tools | $35.91 \pm 3.79$ | 4.71% | $4.63 \pm 0.06$ | 42.43% |
| | PAML_adaptive+MVMoE | $34.92 \pm 3.67$ | 7.33% | $0.69 \pm 0.01$ | 91.43% |
| | PAML_m1+MVMoE | $39.45 \pm 3.60$ | -4.69% | $0.57 \pm 0.01$ | 92.96% |
| | PAML_m3+MVMoE | $35.75 \pm 3.40$ | 5.12% | $0.61 \pm 0.01$ | 92.39% |
| 500 | OR-Tools | $39.64 \pm 3.73$ | 9.38% | $10.07 \pm 0.01$ | - |
| | MVMoE | $43.74 \pm 3.22$ | - | $0.94 \pm 0.01$ | 90.67% |
| | PAML_adaptive+OR-Tools | $39.88 \pm 4.00$ | 8.83% | $7.63 \pm 0.07$ | 24.26% |
| | PAML_m1+OR-Tools | $42.46 \pm 4.26$ | 2.93% | $3.76 \pm 0.03$ | 62.63% |
| | PAML_m3+OR-Tools | $41.02 \pm 4.49$ | 6.22% | $4.79 \pm 0.03$ | 52.46% |
| | PAML_adaptive+MVMoE | $39.95 \pm 4.16$ | 8.67% | $0.87 \pm 0.01$ | 91.35% |
| | PAML_m1+MVMoE | $45.21 \pm 4.54$ | -3.36% | $0.71 \pm 0.01$ | 92.94% |
| | PAML_m3+MVMoE | $40.82 \pm 4.20$ | 6.67% | $0.76 \pm 0.02$ | 92.41% |
| 1000 | OR-Tools | $55.05 \pm 4.59$ | 30.92% | $20.26 \pm 0.00$ | - |
| | MVMoE | $79.69 \pm 6.40$ | - | $4.04 \pm 0.01$ | 80.07% |
| | PAML_adaptive+OR-Tools | $61.40 \pm 5.89$ | 22.96% | $8.35 \pm 0.05$ | 58.79% |
| | PAML_m1+OR-Tools | $73.08 \pm 7.02$ | 8.30% | $3.96 \pm 0.01$ | 80.47% |
| | PAML_m3+OR-Tools | $66.01 \pm 6.86$ | 17.17% | $5.44 \pm 0.06$ | 73.15% |
| | PAML_adaptive+MVMoE | $63.23 \pm 6.29$ | 20.66% | $1.58 \pm 0.07$ | 92.20% |
| | PAML_m1+MVMoE | $80.48 \pm 8.07$ | -0.98% | $1.34 \pm 0.01$ | 93.40% |
| | PAML_m3+MVMoE | $66.07 \pm 6.51$ | 17.09% | $1.42 \pm 0.02$ | 93.01% |
| 2000 | OR-Tools | $82.05 \pm 7.19$ | 65.51% | $41.13 \pm 0.03$ | - |
| | MVMoE | $237.87 \pm 33.08$ | - | $31.18 \pm 0.07$ | 24.21% |
| | PAML_adaptive+OR-Tools | $110.20 \pm 7.83$ | 53.67% | $12.61 \pm 0.02$ | 69.34% |
| | PAML_m1+OR-Tools | $198.02 \pm 34.60$ | 16.75% | $5.00 \pm 0.03$ | 87.85% |
| | PAML_m3+OR-Tools | $147.67 \pm 16.26$ | 37.92% | $7.61 \pm 1.42$ | 81.50% |
| | PAML_adaptive+MVMoE | $122.01 \pm 6.97$ | 48.71% | $3.62 \pm 0.05$ | 91.20% |
| | PAML_m1+MVMoE | $220.02 \pm 37.93$ | 7.51% | $2.59 \pm 0.02$ | 93.71% |
| | PAML_m3+MVMoE | $152.19 \pm 15.66$ | 36.02% | $2.73 \pm 0.05$ | 93.37% |

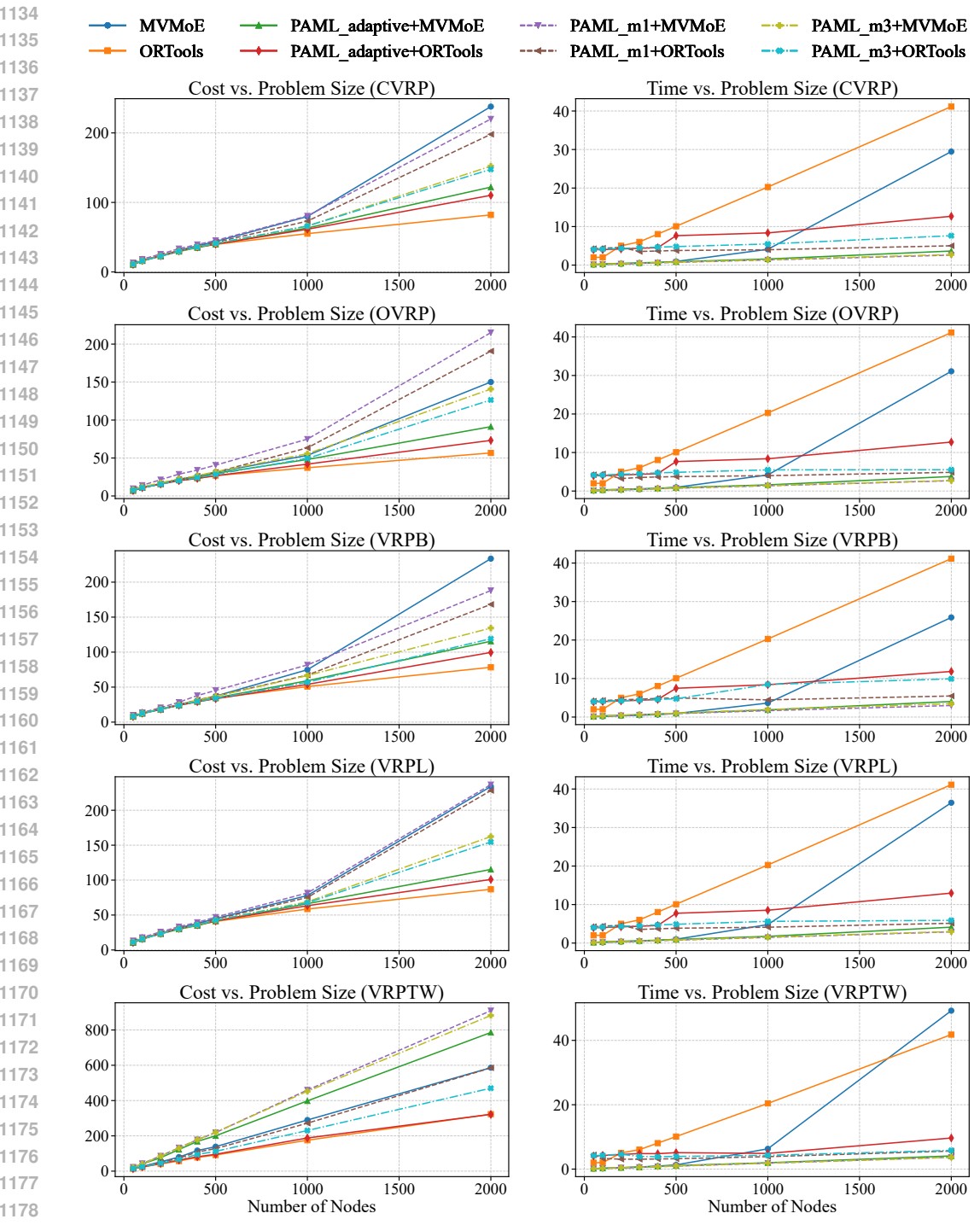

Figure 6: This figure presents a comprehensive evaluation of eight routing algorithms on five vehicle routing problem variants (CVRP, OVRP, VRPB, VRPL, VRPTW) across varying problem sizes (50-2000 nodes). The left column displays solution quality (total cost), while the right column shows computational time.

Table 3 and Figure 6 shows the detailed performance data for solving VRP problems with 50-2000 nodes. Data includes cost (path length), cost improvement relative to MVMoE (C_Imp), solving time, and time improvement relative to OR-Tools (T_Imp).

Table 3 presents the aggregated average results across VRP variants (CVRP, OVRP, VRP and VRPL), providing a comprehensive overview of algorithmic performance. In contrast, Figure 6 offers a detailed breakdown by individual problem variant, enabling variant-specific analysis of solution quality and computational efficiency across different problem scales.

## D.3    RESULTS ON CVRPLIB INSTANCES

Table 4: Solution Results of Different Methods on CVRPLIB Instances

| Instance | Nodes | Metric | Method | | | |
|---|---|---|---|---|---|---|
| | | | MVMoE | OR-Tools | PAML+MVMoE | PAML+OR-Tools |
| X-n106-k14 | 105 | COST | 27018.10 | 27331.98 | 27018.10 | 27331.98 |
| | | TIME | 0.17 | 5.01 | 0.33 | 5.23 |
| X-n167-k10 | 166 | COST | 21043.58 | 22477.14 | 21585.61 | 22216.27 |
| | | TIME | 0.24 | 5.01 | 0.39 | 2.93 |
| X-n214-k11 | 213 | COST | 12080.17 | 12723.81 | 12090.17 | 11821.63 |
| | | TIME | 0.32 | 10.02 | 0.44 | 3.03 |
| X-n298-k31 | 297 | COST | 40047.84 | 40069.33 | 37006.62 | 37829.50 |
| | | TIME | 0.45 | 10.03 | 0.60 | 5.57 |
| X-n322-k28 | 321 | COST | 34865.78 | 33016.60 | 31516.92 | 32115.58 |
| | | TIME | 0.48 | 10.03 | 0.62 | 5.58 |
| X-n351-k40 | 350 | COST | 31838.01 | 29013.45 | 29952.35 | 28453.90 |
| | | TIME | 0.53 | 10.05 | 0.68 | 5.66 |
| X-n429-k61 | 428 | COST | 76237.32 | 69578.19 | 70310.74 | 70009.27 |
| | | TIME | 0.67 | 10.08 | 0.81 | 5.78 |
| X-n491-k59 | 490 | COST | 81748.93 | 74379.78 | 79719.71 | 72891.84 |
| | | TIME | 0.75 | 10.09 | 0.92 | 5.86 |
| X-n561-k42 | 560 | COST | 55257.19 | 47755.19 | 52610.73 | 48270.21 |
| | | TIME | 0.84 | 20.13 | 1.34 | 6.29 |
| X-n613-k62 | 612 | COST | 82537.28 | 68333.87 | 68126.37 | 68192.57 |
| | | TIME | 0.92 | 20.13 | 1.05 | 6.05 |
| X-n685-k75 | 684 | COST | 96661.72 | 80230.82 | 80016.56 | 75389.39 |
| | | TIME | 1.03 | 20.16 | 1.16 | 6.16 |
| X-n766-k71 | 765 | COST | 166986.46 | 125943.27 | 143674.88 | 130772.26 |
| | | TIME | 1.14 | 20.20 | 1.31 | 6.23 |
| X-n783-k48 | 782 | COST | 106156.18 | 80451.99 | 84432.11 | 81113.05 |
| | | TIME | 1.13 | 20.20 | 1.27 | 6.22 |
| X-n856-k95 | 855 | COST | 128456.82 | 91631.11 | 107077.03 | 103600.83 |
| | | TIME | 1.27 | 20.24 | 1.43 | 6.40 |
| X-n895-k37 | 894 | COST | 100030.74 | 59331.74 | 69079.78 | 61944.52 |
| | | TIME | 1.26 | 20.26 | 1.37 | 6.41 |
| X-n957-k87 | 956 | COST | 140497.89 | 88740.07 | 97970.54 | 91847.58 |
| | | TIME | 1.38 | 20.30 | 1.54 | 6.51 |
| Average | - | AVG_COST | 75091.50 | 59438.02 | 63261.76 | 60237.52 |
| | | C_Imp(%) | - | 20.85 | 15.75 | 19.78 |
| | | AVG_TIME | 0.79 | 14.50 | 0.95 | 5.62 |
| | | T_Imp(%) | 94.58 | - | 93.42 | 61.24 |

Table 4, Figure 7 and Figure 8 shows the results of solving CVRPLIB dataset instances using different methods. The data includes the number of nodes for each instance, solution cost, and solving time.

Table 4 presents detailed instance-level results from the standard CVRPLIB benchmark, showcasing the performance of four methods across instances ranging from 105 to 956 nodes. Figure 7 provides a comparative distribution analysis through box plots, illustrating the variability and central tendencies of both solution costs and computation times across methods. And Figure 8 offers statistical validation of performance improvements, with the left panel displaying smoothed cost improvement trends across problem scales and the right panel presenting statistical significance through a heatmap visualization.

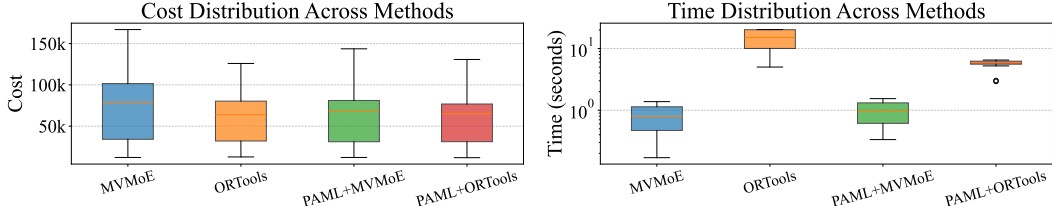

Figure 7: Cost and Time Distribution Across Methods. This figure illustrates the distribution of cost and computation time for the four methods using box plots.

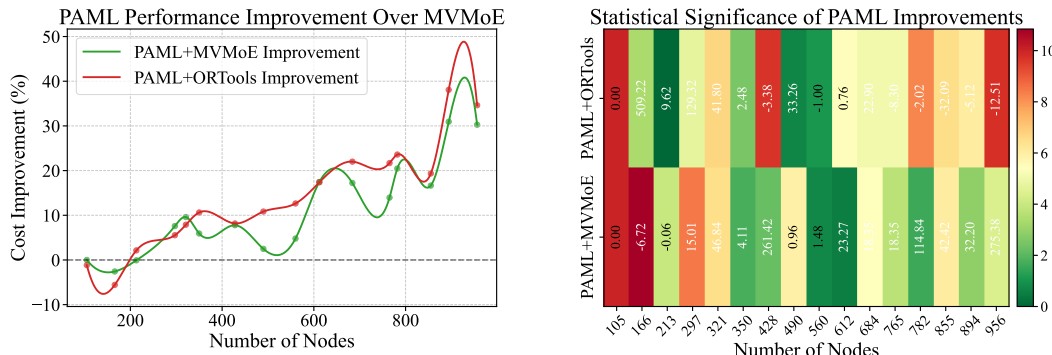

Figure 8: PAML Performance Improvement and Statistical Significance. The left panel shows the cost improvement percentage of PAML+MVMoE and PAML+OR-Tools over their baselines across different problem sizes, smoothed using cubic interpolation. The right panel presents the statistical significance of these improvements using a heatmap, with -log10(p-values) and annotated effect sizes. Significant improvements (p<0.05) are highlighted in white text.

## D.4   VISUALIZATION OF CVRPLIB SOLUTIONS

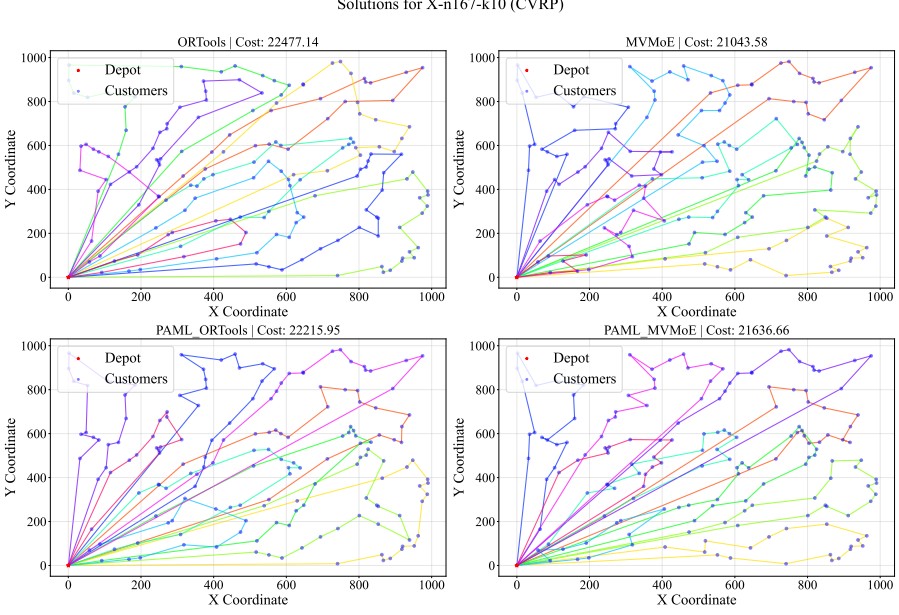

Figure 9: Visualization of solution for X-n106-k14 instance (105 nodes)

Figure 10: Visualization of solution for X-n167-k10 instance (166 nodes)

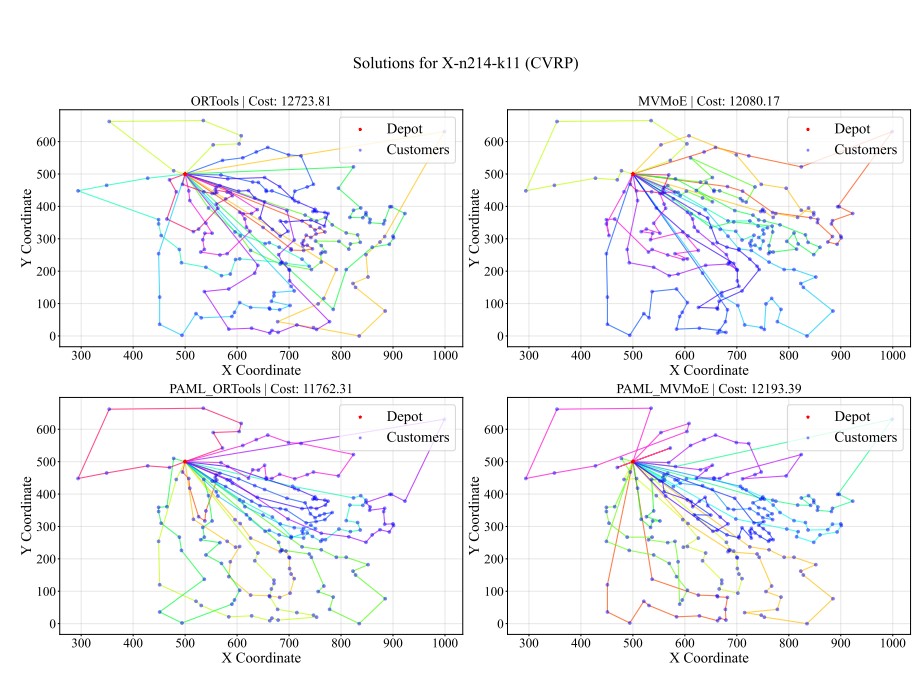

Figure 11: Visualization of solution for X-n214-k11 instance (213 nodes)

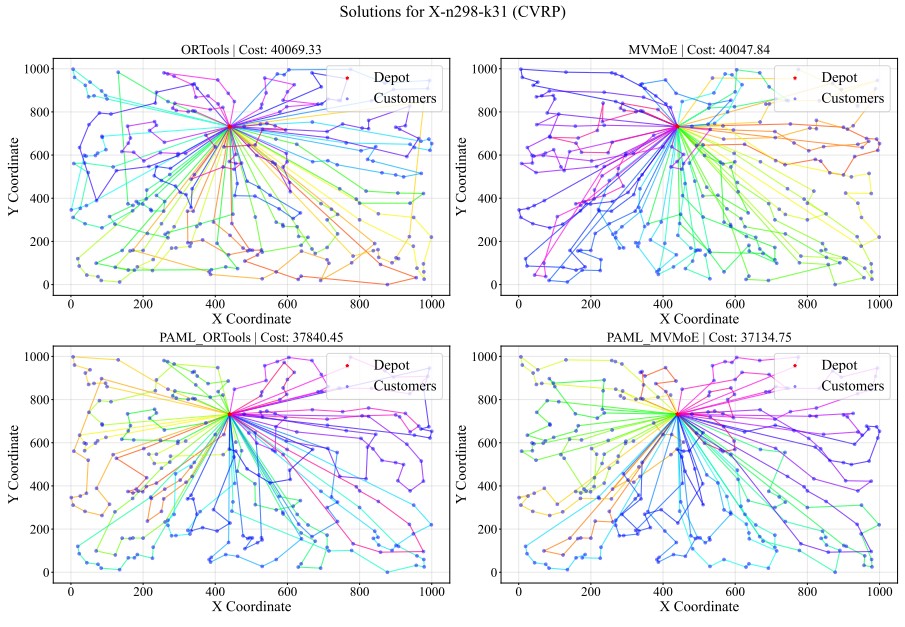

Figure 12: Visualization of solution for X-n298-k31 instance (297 nodes)

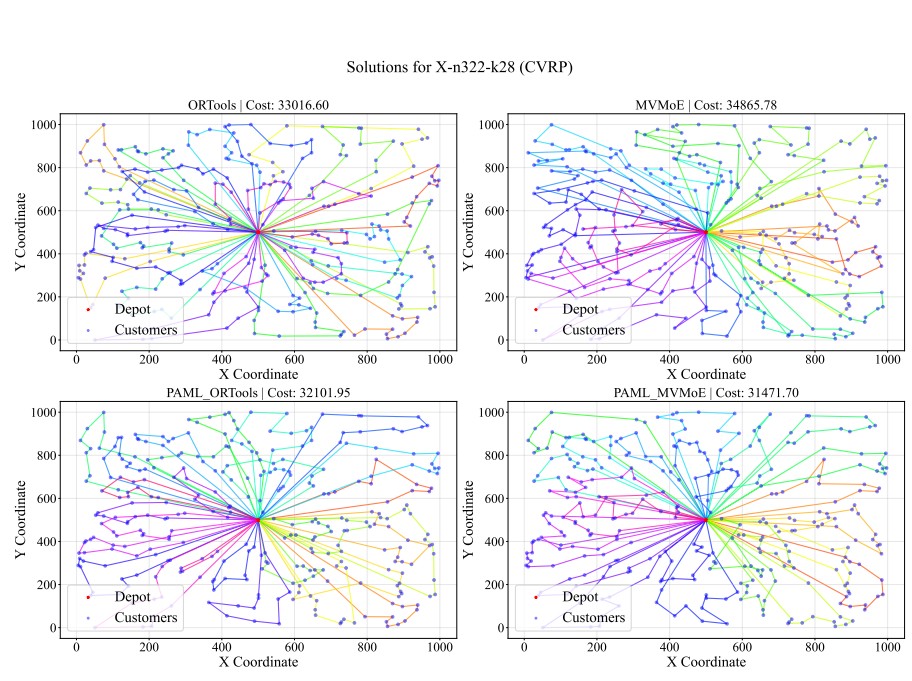

Figure 13: Visualization of solution for X-n322-k28 instance (321 nodes)

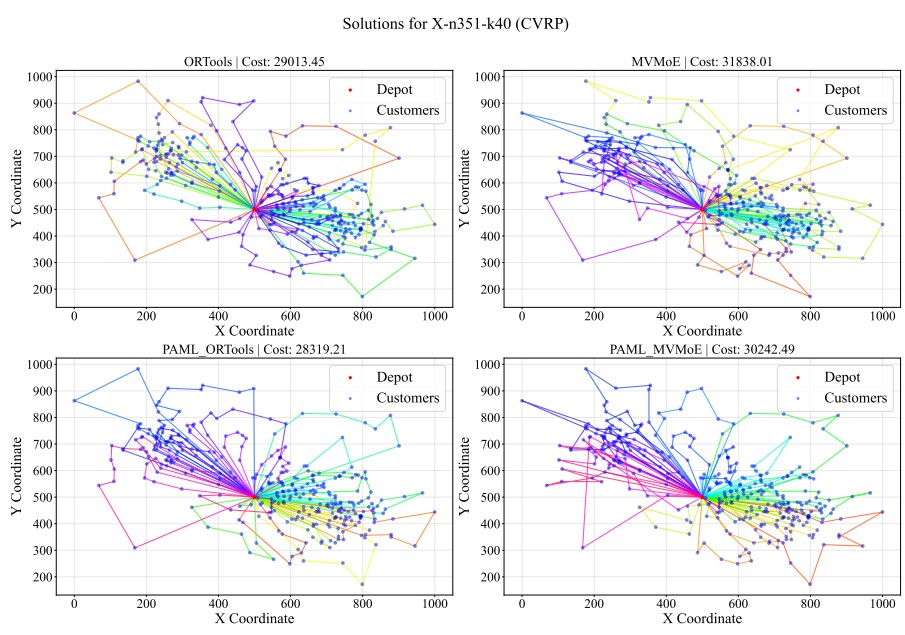

Figure 14: Visualization of solution for X-n351-k40 instance (350 nodes)

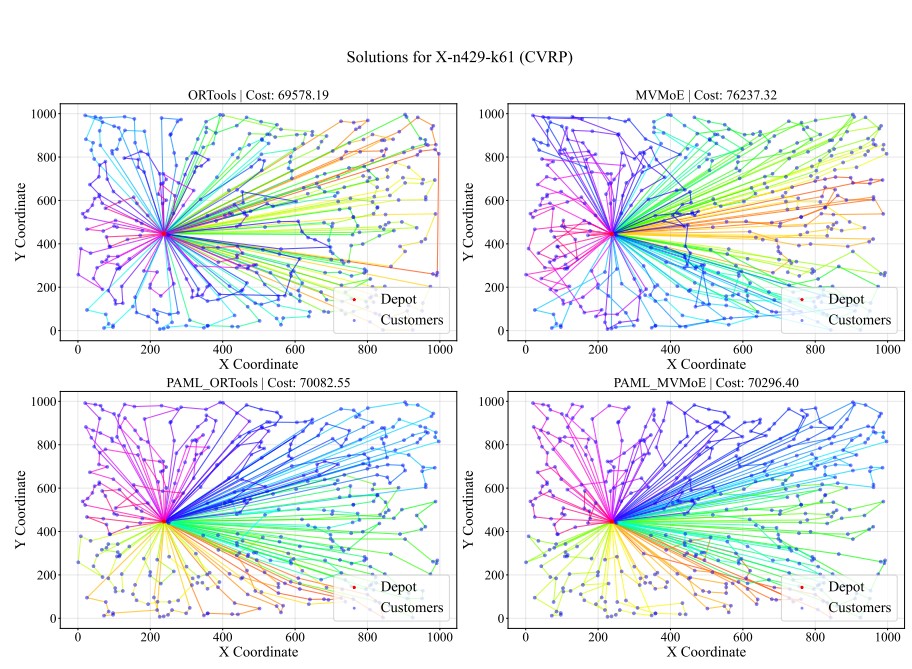

Figure 15: Visualization of solution for X-n429-k61 instance (428 nodes)

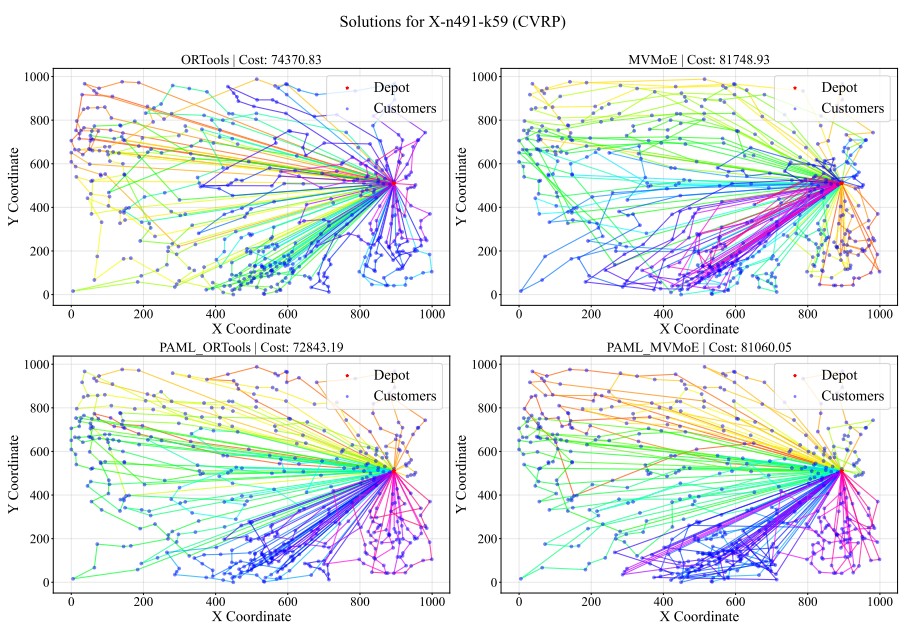

Figure 16: Visualization of solution for X-n491-k59 instance (490 nodes)

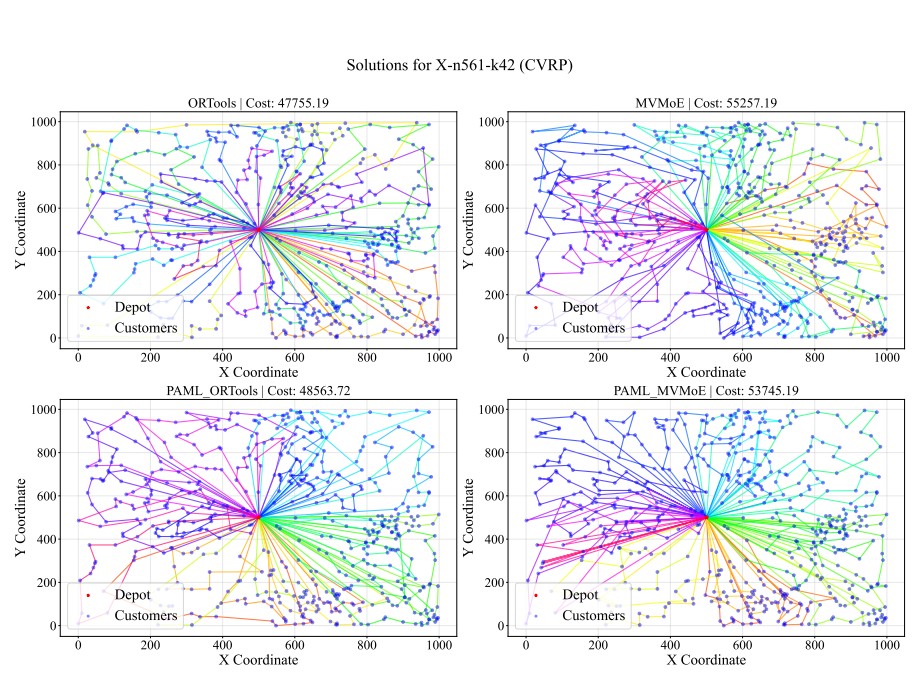

Figure 17: Visualization of solution for X-n561-k42 instance (560 nodes)

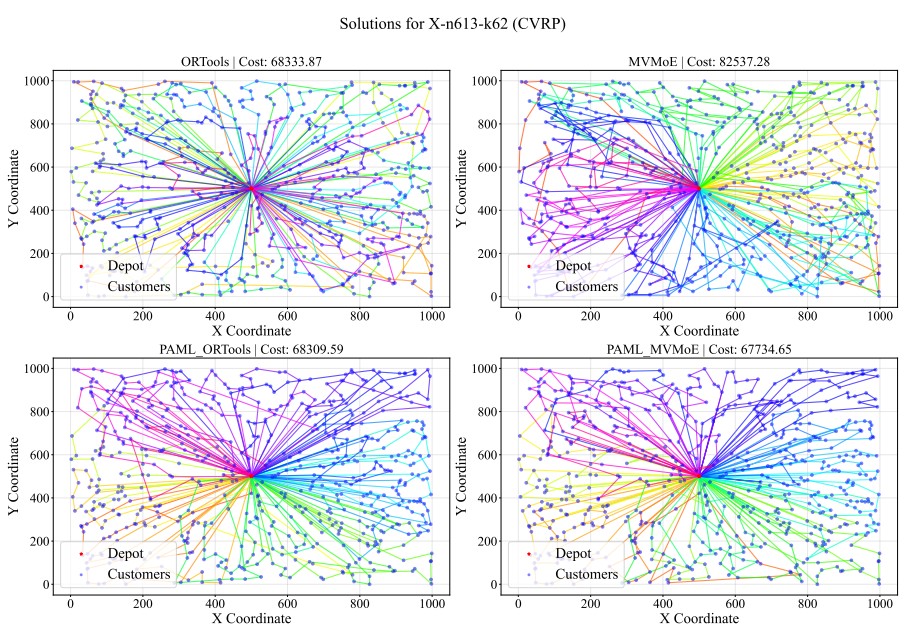

Figure 18: Visualization of solution for X-n613-k62 instance (612 nodes)

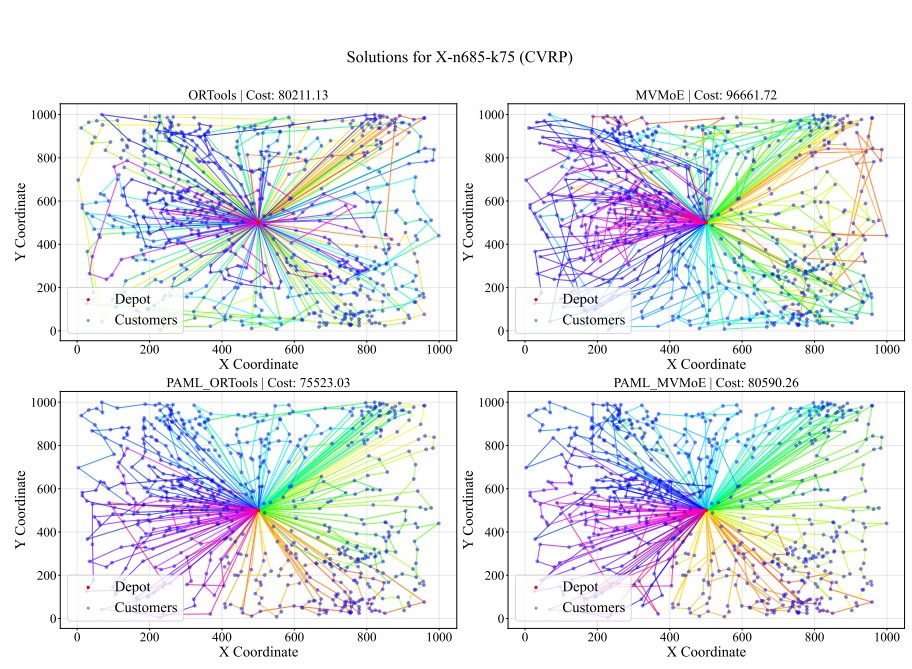

Figure 19: Visualization of solution for X-n685-k75 instance (684 nodes)

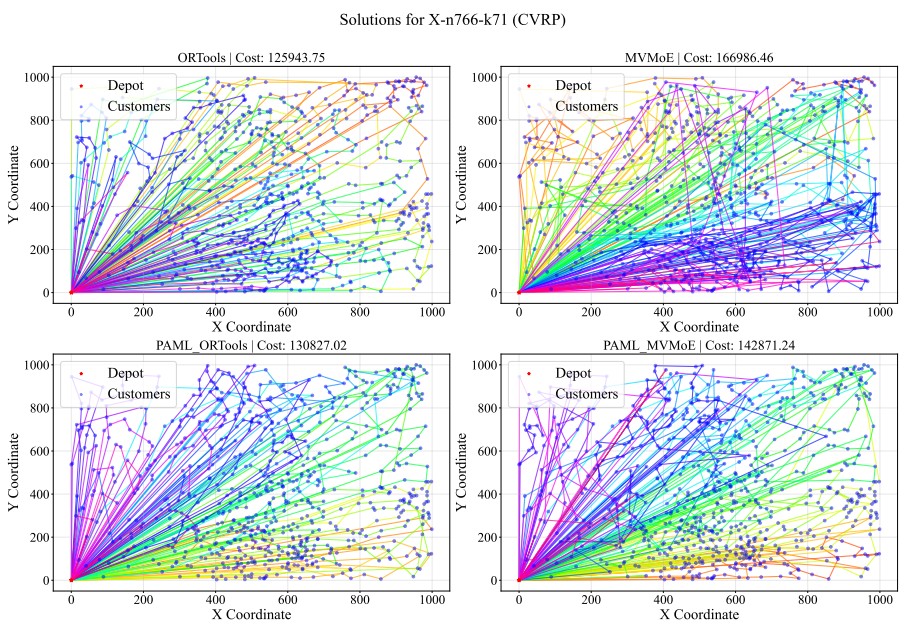

Figure 20: Visualization of solution for X-n766-k71 instance (765 nodes)

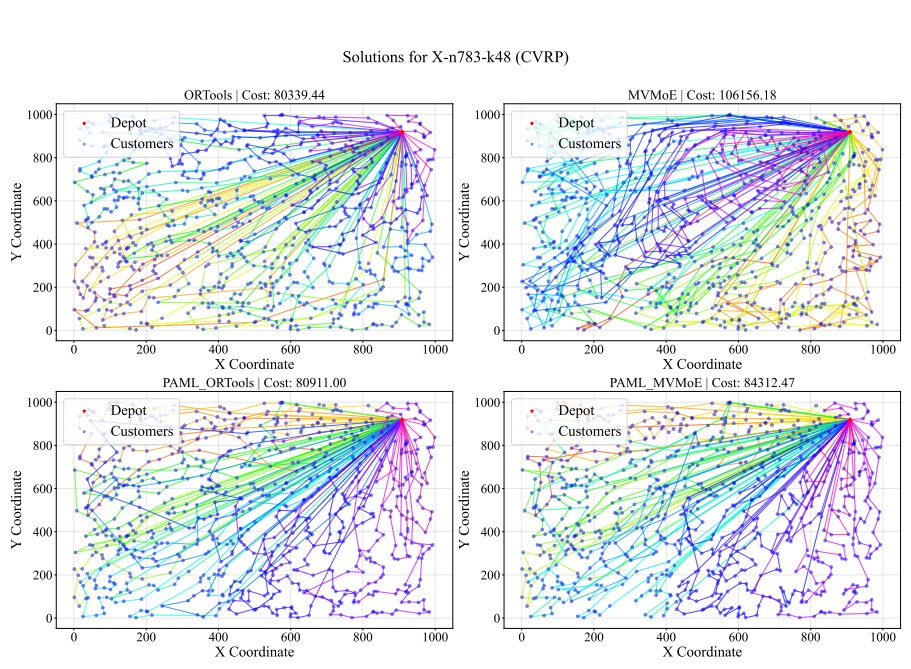

Figure 21: Visualization of solution for X-n783-k48 instance (782 nodes)

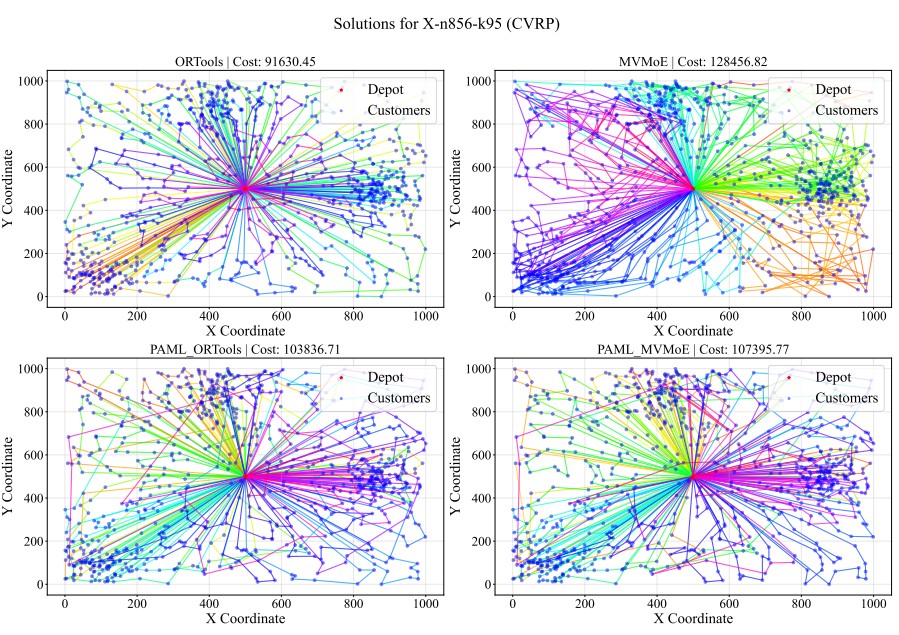

Figure 22: Visualization of solution for X-n856-k95 instance (855 nodes)

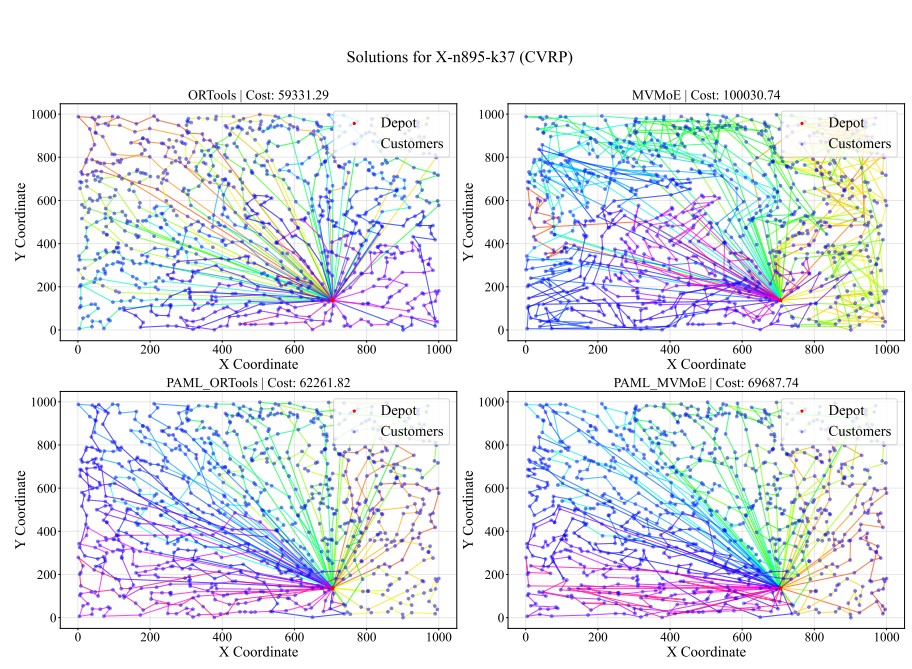

Figure 23: Visualization of solution for X-n895-k37 instance (894 nodes)

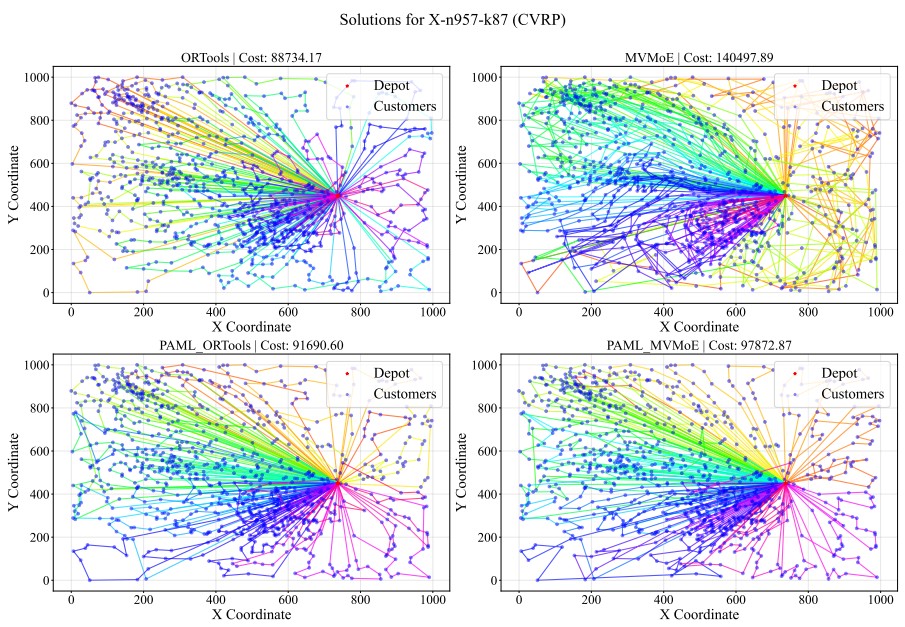

Figure 24: Visualization of solution for X-n957-k87 instance (956 nodes)