# OpenReview forum: "PAML: MoE-Based Partitioning and Merging Framework for Solving Large-scale Multi-task VRPs"
_ICLR.cc/2026/Conference — ICLR 2026 Conference Withdrawn Submission_

### Official Review · Reviewer_iY9t · 2025-10-27

**Soundness:** 2
**Presentation:** 2
**Contribution:** 1
**Rating:** 2
**Confidence:** 5

**Summary:**

This paper presents the PAML framework, a novel partitioning-based method designed to tackle medium- to large-scale Vehicle Routing Problem (VRP) instances and their variants. The core idea of PAML involves three stages: 1) decomposing a large problem into smaller subproblems using a reinforcement learning-based implicit partitioner, 2) dynamically merging these subproblems based on polar angle clustering to optimize their sizes, and 3) solving the merged subproblems in parallel using a multi-task solver. This divide-and-conquer strategy builds on established frameworks like TAM and MVMoE, enhancing scalability and computational efficiency. Experimental results highlight significant improvements in solution quality and efficiency, achieving up to 48.71% reduction in route length for 2000-node problems while maintaining computational performance comparable to industry-standard solvers like OR-Tools.

**Strengths:**

This is a complete paper. The authors attempt to convey the main ideas and clearly state the advantages of their proposed method. For medium-scale VRPs and their variants, the method achieves improvements in both speed and solution quality.

**Weaknesses:**

1. The paper's innovation is somewhat lacking. While it utilizes existing decomposition models and multi-task solvers like MVMoE, it fails to introduce significant novel elements or methodologies to genuinely advance the field.

2. The appendix results reveal that the method's performance on 1000- and 2000-node instances is underwhelming, indicating a notable gap between the paper's theoretical promises and its practical effectiveness.

3. The methodology section is encumbered by redundant formulas and lacks clarity in explaining key processes. The end-to-end training of the partitioner, subproblem merging, and final solution combination are not adequately detailed. The appendix also fails to provide vivid or comprehensive explanations, relying too much on text without visual aids. A more concise and clear description of these processes in the main text is essential to enhance the paper's technical validity.

4. The figures in the paper are repetitive and do not add significant value. Figures 1 and 2 illustrate the same workflow without offering complementary insights, and they are not referenced or explained in the main text, which reduces their utility.

5. The experimental section has multiple issues. The descriptions of the generated datasets and real-world benchmarks are vague, making it hard to evaluate the testing scenarios. The lack of detailed numerical tables obscures the model's specific performance and the experimental design's validity. Additionally, the baselines are too limited, only including OR-Tools and MVMoE, which restricts a comprehensive assessment of the model's capabilities.

6. The ablation study and Table 1 do not significantly contribute to the paper's core narrative. Removing them and including more main experimental results would make the paper more convincing.

**Questions:**

1. Is the method capable of efficiently solving problems with 10k nodes or more, and if so, what adaptations ensure its effectiveness for truly large-scale scenarios?

2. To enhance reproducibility and understanding, could the authors provide a more detailed breakdown of the algorithmic workflow, including the rationale behind each stage of the framework?

3. To strengthen the experimental evaluation, could the authors include a broader set of baseline methods and provide more granular numerical results, potentially expanding the comparison to include additional solvers beyond OR-Tools?

4. Could the authors elaborate on the specific merging methodology used and justify its effectiveness? Additionally, how does merging prior to solving compare to an alternative approach where solutions are merged after parallel processing?

5. Could the authors provide more details on the implementation of parallel solving and quantify the resulting efficiency improvements?

---

### Official Review · Reviewer_y3JS · 2025-10-29

**Soundness:** 3
**Presentation:** 3
**Contribution:** 3
**Rating:** 2
**Confidence:** 4

**Summary:**

This paper proposes PAML, a three-stage framework that uses an MoE-based learned partitioner to decompose large-scale VRPs into smaller subproblems, applies geometric heuristic merging, and solves subproblems in parallel, achieving up to 48.71% route length reduction on 2000-node problems and 93.32% time reduction on real-world instances compared to baselines (OR-Tools).

**Strengths:**

1. Comprehensive experimental scope: The paper evaluates across multiple VRP variants (CVRP, OVRP, VRPB, VRPL, VRPTW) and scales (50-2000 nodes), including both synthetic and real-world CVRPLIB instances. This breadth demonstrates practical applicability across diverse problem settings.
2. Significant computational efficiency gains: PAML achieves 10× speedup over OR-Tools for 200-node problems and 93.32% time reduction (0.95s vs 14.23s) on CVRPLIB while maintaining competitive solution quality. This computational efficiency is practically valuable for real-time logistics applications.

**Weaknesses:**

1) Training Setups: The training details are not clear (especially the problem sizes used, training cost, etc), which makes it hard to judge the fairness of comparison to baselines and assess the overall computational trade-offs.

2) Technical novelty: The core decomposition strategy appears to be adapted from TAM (2023), the MoE architecture from MVMoE (2024), and the merging procedure relies on standard geometric heuristics. The paper should clearly articulate what is genuinely novel beyond the combination of these existing techniques, and provide stronger motivation for the design choices. Furthermore, since the subproblem-solving capability depends entirely on pre-trained multi-task VRP solvers (e.g., MVMoE), the performance of PAML is inherently constrained by the underlying solver’s capacity.

3) Presentation quality issues: The paper suffers from important main results demonstration, poor figure quality (Figure 5 has poor readability with lines overlapping), and technical justification missing from main text, which are the major concerns of the presentation quality of whether the paper should be accepted. For example, there are missing justification for the important formula (8b) in section 4.2.5 and long-winded explanations in section 5.1.1 and conclusions (which could be improved by removing the detailed generation methods to the appendix, and refining the conclusion instead of repeating the contributions with much details. Adding limitations analysis or future works are also improvements). More importantly, the main results should be included in the main text (like figure 6 and table 3), those are the critical parts that judges the contributions of the proposed method.

**Questions:**

1) It is unclear to me what subproblem merging method has been identified in “finally identified subproblem merging methods for various multi-task VRPs of various sizes”. Is this a specific named method or just the general approach of using polar angle sorting with adaptive target sizes from Table 1?  This could be explained better.

2) In eqn (6), the author uses the concatenation of the best found solution to get an optimized solution for the complete problem. Can you explain why concatenating the optimal solutions of subproblems can optimize the objective solution of the complete problem? VRP is a global optimization problem where local optima in subproblems don't guarantee global optimality. How does your partitioning and merging naturally divide good clusters of subroutes that lead to near-optimal complete solutions?

3) Why use this LMoE gating mechanism (in eqn (9)) for MOE gating instead of other MOE gating variants like expert choice or soft gating? An appropriate reference to such a mechanism and justification is could improve the methodology presentation.

4)The training setups seem to be missing in the main manuscript or lack a reference to the Appendix. Without those details, it is difficult to evaluate the reproducibility of the proposed method. Could you add a training configuration table in the main text or provide a clear reference to the appendix section containing these details?

5) How does PAML guarantee constraint satisfaction (capacity, time windows, route length) when merging subproblems? The penalty terms in Equation 8b (V_cap, V_tw, V_route) suggest soft constraints during training—what happens if merged subproblems become infeasible during inference? Do you verify feasibility before passing to the solver, and if so, how do you handle infeasible merges?

6) Can you compare the proposed method with other SOTA MTL solvers? (e.g.MTPOMO, ReLD-MTL, GOAL-MTL)

---

### Official Review · Reviewer_pHsC · 2025-10-30

**Soundness:** 2
**Presentation:** 2
**Contribution:** 2
**Rating:** 2
**Confidence:** 3

**Summary:**

This paper addresses the challenge of scaling Vehicle Routing Problem (VRP) solvers to large instances by introducing a decomposition-based framework. The proposed method, MoE-based Partitioning and Merging for Large-scale VRP (PAML), decomposes large VRPs into sub-routes, merges these into subproblems, solves them in parallel, and then combines the resulting solutions to form a complete solution to the original problem.
The authors validate the effectiveness of PAML using two baseline solvers — the neural solver MvMoE and the heuristic solver OR-Tools — on both synthetic and real datasets (CVRPLIB). Empirical results show that applying PAML leads to improved objective values compared to the baselines without partitioning.

**Strengths:**

- The proposed framework is compatible with existing heuristic solvers, and empirical results demonstrate its effectiveness when combined with both learning-based and classical solvers.

- The experimental evaluation covers multiple VRP variants, suggesting that the framework can accommodate diverse constraint types across different VRP formulations.

**Weaknesses:**

- Although the paper focuses on problem decomposition, it does not compare the proposed framework with other decomposition approaches. Thus, it remains unclear whether the observed improvements stem from the specific MoE-based partitioning design or simply from performing any reasonable decomposition. Since large-scale VRP decomposition is a common idea, comparisons with simpler baselines (e.g., clustering-based partitioning or random segmentation) are essential.

- The empirical evaluation does not quantitatively establish why or how much the proposed method outperforms existing partitioning techniques. Without such evidence, it is difficult to assess the real advantage of PAML over simpler or previously established methods.

**Questions:**

- Why, how, and to what extent does the proposed partitioning and merging framework outperform existing problem decomposition approaches?
- What are the characteristics of VRP solvers that make them suitable for integration with PAML?

---

### Official Review · Reviewer_pmv7 · 2025-10-31

**Soundness:** 2
**Presentation:** 2
**Contribution:** 2
**Rating:** 2
**Confidence:** 5

**Summary:**

This paper proposes a MoE-based PAML framework to solve large-scale multi-task Vehicle Routing Problems. It addresses the limitations of existing neural heuristic algorithms by introducing a three-stage pipeline: (1) a learnable implicit partitioner that decomposes large VRPs into smaller sub-routes while preserving constraints; (2) a dynamic heuristic merging mechanism that merges sub-routes into optimal-sized sub-problems using polar or centroid-based clustering; (3) parallel solving of sub-problems using solvers such as MVMOE or ORTools. While the work has some merit, the main contribution does not meet the novelty expectations of ICLR.

**Strengths:**

1.	The authors propose a merging strategy that can combine multiple sub-routes into a single sub-problem. Compared with TAM, it better preserves global information while balancing the complexity of solving it.

**Weaknesses:**

1.	The core components directly draw from TAM [1] and MVMoE [2], with the main contribution being the proposal of merging integration, which shows limited innovation.
2.	The paper emphasizes the use of a MoE-based partitioner, but does not provide specific details on the expert routing/gating mechanism during the partitioning process (e.g., how experts handle variant constraints). The appendix also does not elaborate on the expert design or routing logic.
3.	The merging motivation is clear, but there is no direct comparison between "partitioning + merging" and end-to-end large subproblem generation (which seems more direct and general).
4.	There is no comparison with recent SOTA methods, such as CaDA [3], etc.
5.	Appendix B.3 presents the multi-task loss function of MVMoE, but in reality, MVMoE trains one task per batch and performs backpropagation to compute gradients, without involving a multi-task loss. Moreover, MVMoE is not built on the platform of Berto et al [4]. (lines 185-188).
6.	There is an error in the equation numbering in Section 4.1.

[1] Generalize learned heuristics to solve large-scale vehicle routing problems in real-time. ICLR 2023.

[2] MVMoE: Multi-Task Vehicle Routing Solver with Mixture-of-Experts. ICML 2024.

[3] Cross-Problem Routing Solver with Constraint-Aware Dual-Attention. ICML 2025.

[4] Routefinder: Towards foundation models for vehicle routing problems. URL https://arxiv.org/abs/2406.15007.

**Questions:**

See the weaknesses.

---

### Note · Authors · 2025-11-23

I have read and agree with the venue's withdrawal policy on behalf of myself and my co-authors.